# Modular detergents tailor the purification and structural analysis of membrane proteins including G-protein coupled receptors

Leonhard H. Urner [1,2,5], Idlir Liko[2,3,5], Hsin-Yung Yen[2,3], Kin-Kuan Hoi[2], Jani Reddy Bolla [2], Joseph Gault [2], Fernando Gonçalves Almeida[3], Marc-Philip Schweder[1], Denis Shutin[2], Svenja Ehrmann[1], Rainer Haag[1*], Carol V. Robinson [2*] & Kevin Pagel[1,4*]

Detergents enable the purification of membrane proteins and are indispensable reagents in structural biology. Even though a large variety of detergents have been developed in the last century, the challenge remains to identify guidelines that allow fine-tuning of detergents for individual applications in membrane protein research. Addressing this challenge, here we introduce the family of oligoglycerol detergents (OGDs). Native mass spectrometry (MS) reveals that the modular OGD architecture offers the ability to control protein purification and to preserve interactions with native membrane lipids during purification. In addition to a broad range of bacterial membrane proteins, OGDs also enable the purification and analysis of a functional G-protein coupled receptor (GPCR). Moreover, given the modular design of these detergents, we anticipate fine-tuning of their properties for specific applications in structural biology. Seen from a broader perspective, this represents a significant advance for the investigation of membrane proteins and their interactions with lipids.

[1] Institute of Chemistry and Biochemistry, Freie Universität Berlin, 14195 Berlin, Germany. [2] Physical and Theoretical Chemistry Laboratory, University of Oxford, Oxford OX1 3QZ, UK. [3] OMass Therapeutics, The Schrödinger Building, Heatley Road, The Oxford Science Park, Oxford OX4 4GE, UK. [4] Department of Molecular Physics, Fritz Haber Institute of the Max Planck Society, 14195 Berlin, Germany. [5]These authors contributed equally: Leonhard H. Urner, Idlir Liko. *email: haag@zedat.fu-berlin.de; carol.robinson@chem.ox.ac.uk; kevin.pagel@fu-berlin.de

Membrane proteins are targets for more than 50% of current drugs[1]. Among them, GPCRs are the most intensively studied due to their substantial role in human health and disease[2]. At the same time, investigating bacterial membrane proteins is key for improvements in the development of antibiotics[3]. The native host environment of all membrane proteins is, however, highly dynamic and heterogeneous and this constitutes a bottleneck for their direct structural analysis. Consequently, membrane protein complexes are extracted from their native environment with detergents, which are traditionally used to dissolve biological membranes. Detergents disrupt lipid–lipid and protein–lipid interactions in membranes and form soluble proteomicelles by shielding the hydrophobic surfaces of proteins from water[4–7]. Proteomicelles enable the purification of membrane proteins by separation techniques and in this way detergents facilitate the structural elucidation of this unique protein class.

Although the field of synthetic detergent chemistry is well established, with first reports more than 100 years ago[8], suitable detergents for membrane protein research are still typically identified by trial and error. The number of detergent families is staggeringly diverse and their chemical variety makes it increasingly difficult to define design guidelines to predict the utility of detergents for purifying membrane proteins. Furthermore, the concentration of detergent used in purification protocols is often adjusted to a multiple of its' critical aggregation concentration (cac)[9]. Interestingly, however, detergents with similar cac values can exhibit opposing compatibilities with proteins. For example, sodium dodecyl sulfate (SDS) and tetraethylene glycol monooctyl ether (C8E4) have similar cac values. However, SDS is a strong protein denaturant while C8E4 preserves native structural features of various membrane proteins. This makes it very difficult to predict the utility of detergents for protein purification according to their cac values. Together these attributes contribute to the emergence of the universal dogma that the selection of detergents depends more on empirical factors than on scientific principles[10,11].

Here we investigate whether or not the ability of detergents to purify or analyze membrane proteins can be optimized by changing their molecular structure. To address this question, we introduce the family of oligoglycerol detergents (OGDs) for membrane protein research. Research on this family of detergents has primarily been focused on the structure-based understanding of their self-assembly in aqueous solution and the potential of their aggregates to be used as nanocarriers for hydrophobic drugs[12–15]. The modular architecture of OGDs, and underlying synthetic protocols, allow us to readily fine-tune the structure of the head group, linker, and tail for individual research purposes, which is a valuable perquisite for bottom-up investigations (Fig. 1). Here, we explore the utility of OGDs for protein purification and apply native mass spectrometry (MS) to study the structure of purified membrane protein complexes after removing the OGD micelle within a mass spectrometer[16]. Recent breakthroughs in high resolution MS technology enable not only the analysis of intact oligomers but also their binding to native membrane lipids[17,18]. Our results provide direct evidence that the modular OGD architecture can be optimized for the isolation of membrane proteins as well as the preservation of protein subunit interactions and binding to native membrane lipids. Our data also show that OGDs enable the purification of a functional neurotensin receptor type 1 (NTSR1)—a member of the GPCR family, which is currently one of the most challenging and interesting protein classes in pharmacology[2].

## Results

**Rational for OGD design.** The ability to tune a particular detergent for isolating large protein quantities, and at the same time preserving protein interactions to native membrane lipids during isolation, is a major goal in membrane protein research[4–6].

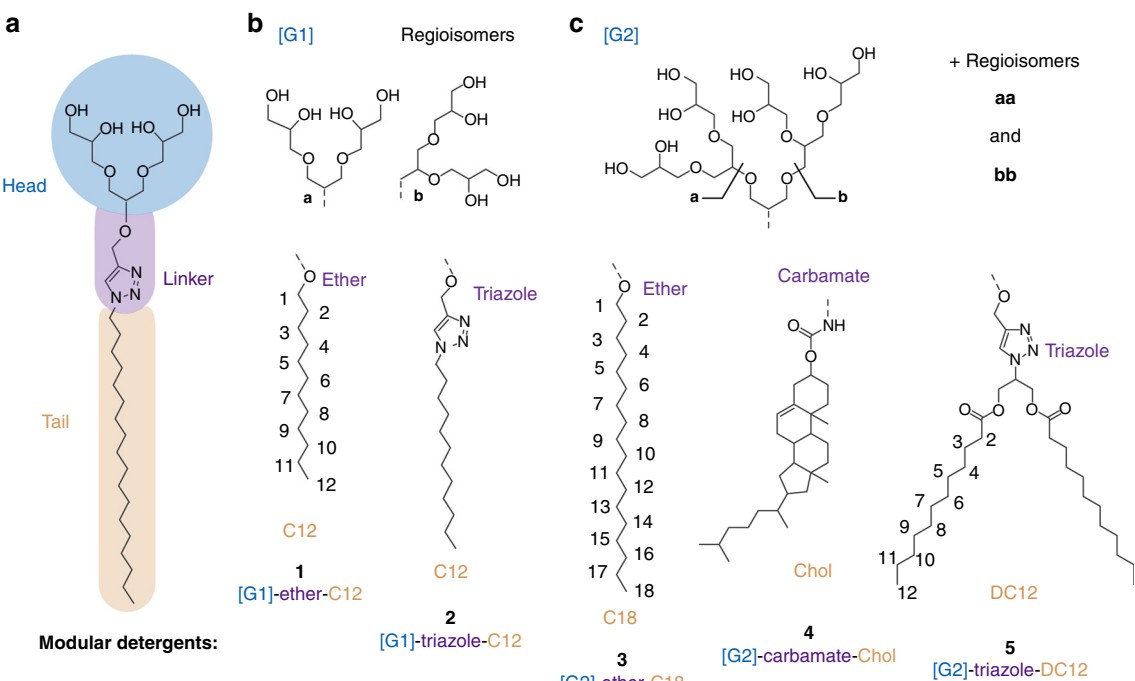

**Fig. 1 Describing the modular architecture of OGDs. a** The molecular architecture of OGDs comprises a hydrophilic head, a hydrophobic tail, and a connecting linker. **b**, **c** OGD regioisomer mixtures based on first-generation [G1] or second-generation [G2] triglycerol are composed of different head group regioisomers (top), linker, and tail structures (bottom). The modular OGD architecture allows this detergent family to be optimized for protein purification, charge reduction, and lipid co-purification.

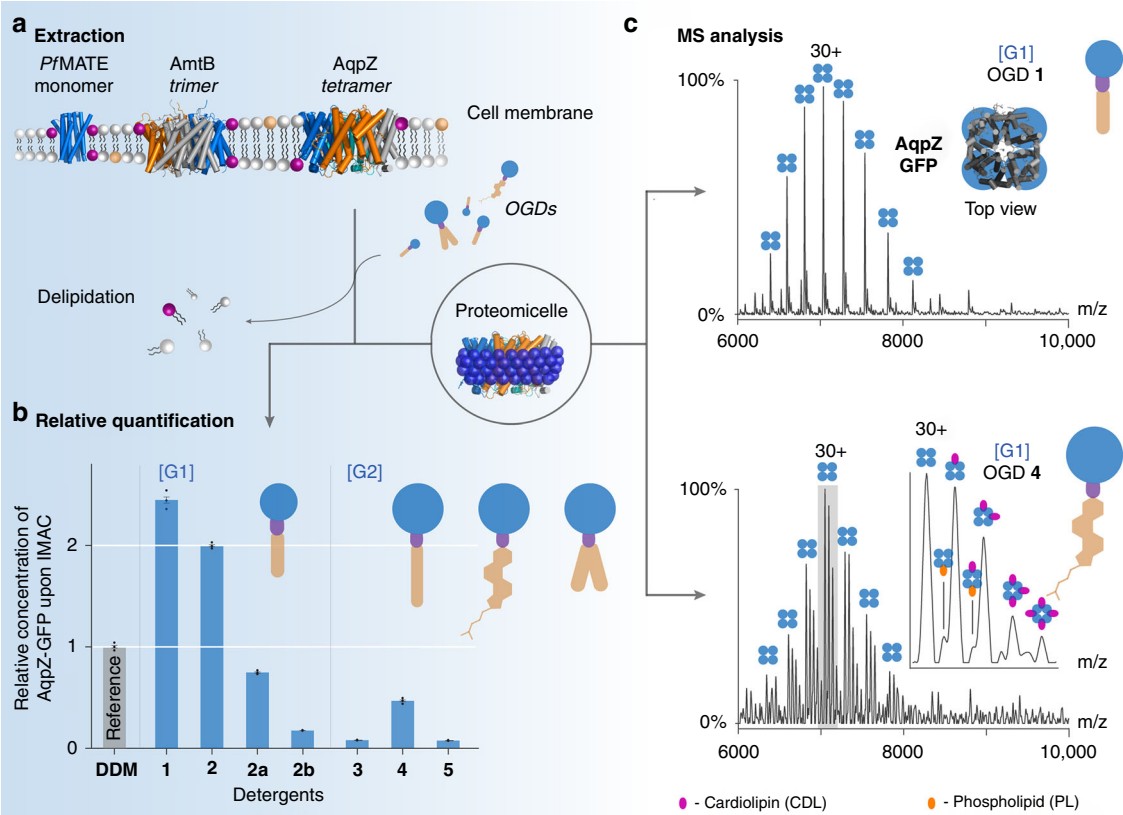

**Fig. 2 OGDs tailor the purification and native MS analysis of membrane proteins. a** Three membrane proteins were isolated from cell membranes using OGDs **1**–**5**. **b** Higher relative protein concentrations are obtained from [G1] OGDs. The [G1] OGD regioisomer mixture **2** (=**2a** + **2b**) leads to a higher extraction yield than the individual regioisomers **2a** and **2b**. Relative protein concentrations were plotted with standard deviation (± s.d., n = 3). **c** Mass spectra obtained from tetrameric AqpZ-GFP after isolation with the [G1] OGD regioisomer mixture **1** reveal no protein complexes with native membrane lipids (upper spectrum). Substitution of the head group and hydrophobic tail in the [G2] OGD regioisomer mixture **4** enabled the detection of protein complexes with native membrane lipids (lower spectrum). Spectra were acquired using similar instrument conditions (HCD energy: 200 V). AqpZ tetramers are indicated by four blue circles, cardiolipins by violet circles, and phospholipids by orange circles. Source data are provided as a Source Data file.

Although proteomicelles provide the water-solubility that is necessary for protein purification, they poorly mimic the heterogeneous structure of lipid membranes[19]. This causes protein instability and, in the worst case, it reduces the amounts of protein that can be isolated.

We hypothesized that increased heterogeneity of OGDs might enhance their ability to replicate natural membrane environments. For this purpose, we synthesized detergent regioisomer mixtures based on first [G1] and second [G2] generation dendritic triglycerol (**1**–**5**, Fig. 1). Previous reports have shown that the size of the detergent head is inversely correlated to the ability of a particular detergent to dissolve lipid membranes[4]. Specifically, the larger the detergent head, the lower the detergent's propensity to break up lipid–lipid and protein–lipid interactions in biological membranes. Based on this result, we anticipate that tuning the size of the OGD head, e.g. [G1] or [G2], will have a direct impact on the propensity of OGDs to extract and delipidate membrane proteins. Furthermore, the high water-solubility of the [G2] head also enables the synthesis of water-soluble OGDs using lipid-solubilizing cholesterol or double chain motifs, which closely resemble the structure of biological phospholipids. In addition, we reasoned that changing the basicity of the linker between the OGD head and tail, e.g. ether/carbamate < triazole, would allow us to effectively tune charge states of membrane proteins in mass spectra—a feature that is of enormous interest within the emerging native MS community[20–24]. Having synthesized these detergents we now

evaluate their ability to extract membrane proteins, retain subunit interactions, and lipid binding properties, as well as to produce favorable characteristics for native mass spectra.

**OGD design and protein isolation**. First, we probed the utility of the OGD design for membrane protein isolation. Different constructs of the aquaporin channel (AqpZ), the ammonia channel (AmtB), and the multidrug efflux pump (MATE) were extracted from *E. coli* membranes using **1**–**5** (Fig. 2a and Supplementary Figs. 1 and 2). Following a previous purification protocol[25], cell membranes were solubilized for 16 h and purified via immobilized metal ion affinity chromatography (IMAC). The relative protein amounts were determined by UV/VIS spectroscopy. Subsequently, the relative protein amounts obtained from **1**–**5** were compared with *n*-dodecyl-ß-D-maltoside (DDM). This detergent enables the purification of high protein yields and is a current standard in structural biology (Fig. 2b)[26].

In the case of AqpZ, we found that the relative protein concentrations obtained following extraction and purification with **1** and **2** were about two times higher than with DDM (Fig. 2b). This highlights the potential of [G1] OGD regioisomer mixtures for isolating high yields of proteins from cell membranes. Interestingly, protein quantities obtained from **2**, which contained two OGD regioisomers, were higher than that of the individual regioisomers **2a** and **2b**. We found similar trends in the case of AmtB (Supplementary Fig. 1). However, differences

in extraction efficiency between the regioisomer mixture **2** and the individual isomers **2a** and **2b** were less pronounced than in the case of AqpZ, but still statistically significant (Supplementary Table 1). To the best of our knowledge, such an increase in solubilized protein quantities arising from a mixture of detergent regioisomers has not been reported previously. In contrast, the relative protein concentrations of AqpZ obtained from [G2] OGD regioisomer mixtures **3–5** were lower than those obtained from [G1] OGD regioisomer mixtures **1–2** or DDM. Similar results were obtained for the isolation of AmtB and MATE (Supplementary Figs. 1 and 2).

To assess the secondary structure of the proteins upon purification we investigated the samples using circular dichroism (CD) spectroscopy. A high alpha-helical content was observed in samples that were isolated by DDM as well as by the [G1] and [G2] regioisomer mixtures **1** and **4** (Supplementary Fig. 3). This observation underlines the utility of OGDs for preserving the native secondary structure of alpha-helical membrane proteins during purification. In addition, we noticed high beta-sheet content in the case of GFP-tagged proteins, such as MATE and AqpZ, which shows that the native secondary structure of the tags was also preserved.

Apart from isolating alpha-helical membrane proteins from membranes, such as MATE, AqpZ, and AmtB, we also attempted to refold and solubilize the G236K/K237G mutant of the beta-barrel outer membrane protein T (OmpT) with DDM as well as with OGD regioisomer mixtures. First, we solubilized OmpT from inclusion bodies with urea, then diluted the OmpT-urea mixture into detergent-containing refolding buffer, and isolated the protein using IMAC. Again higher relative protein quantities were obtained from DDM and the [G1] OGD regioisomer mixture **1**. Analysis of the ratio between folded and unfolded OmpT revealed higher relative proportions of folded OmpT in the case of DDM (Supplementary Fig. 4). However, protein quantities obtained from the [G1] OGD regioisomer mixture **1** were about three times higher than from DDM, thus leading to a higher yield of refolded OmpT under comparable conditions (Supplementary Fig. 4). To investigate if DDM and the [G1] OGD regioisomer mixture **1** can be used to successfully refold OmpT into a functionally active conformation, we enriched folded OmpT to equal amounts using a second IMAC purification step. The mixture of folded and unfolded OmpT was bound to an IMAC column and eluted using an intermediate imidazole concentration (40 mM). This led to an enrichment of folded OmpT to ~70% in both detergent environments (Supplementary Fig. 4). CD spectroscopy revealed a similar high beta-sheet content in both samples, thus confirming the expected secondary structure of OmpT (Supplementary Fig. 4). OmpT is a protease and its proteolytic activity depends on binding to an outer membrane lipid component: smooth lipopolysaccharide (S-LPS)[27]. Time-dependent fluorescence spectroscopy experiments revealed a similar proteolytic activity of refolded OmpT upon addition of S-LPS and the self-quenching fluorescent peptide Abz-ARRAY-Tyr(NO$_2$)-NH$_2$ (Supplementary Fig. 4). As expected, the proteolytic activity of OmpT was enhanced in the presence of S-LPS and reduced in the absence of S-LPS, which underlines that the LPS-mediated protease mechanism is acting in both detergent environments. Our data therefore highlight the utility of DDM and the [G1] OGD regioisomer mixture **1** to solubilize and refold the beta-barrel OmpT into a functionally active conformation.

To further emphasize the general utility of OGDs for the purification of beta-barrel outer membrane proteins, we also attempted to extract the beta-barrel assembly machinery (BAM) from membranes using DDM as well as the [G1] and [G2] OGD regioisomer mixtures **1** and **4**. The BAM complex is responsible for the folding and insertion of beta-barrel proteins into the outer membrane of Gram-negative bacteria and consists of five subunits, including BamA, BamB, BamC, BamD, and BamE[28]. Higher relative protein quantities were obtained upon extraction with DDM and the [G1] OGD regioisomer mixture **1** (Supplementary Fig. 5). Only BamE was overexpressed with a polyhistidine tag. The finding that all five subunits were co-purified during IMAC therefore underlines that DDM and the [G1] OGD regioisomer mixture **1** can extract not only large quantities of the BAM complex, but also preserve subunit interactions during purification. This is an important perquisite for structural studies (Supplementary Fig. 5).

Taken together, OGDs enable the extraction of membrane proteins from cell membranes and also the refolding of membrane proteins. OGD batches containing smaller head groups, such as [G1] OGD regioisomer mixtures **1–2**, led to higher protein quantities than OGD batches with larger head groups, such as [G2] OGD regioisomer mixtures **3–5**. Furthermore, protein quantities obtained among [G2] OGD batches vary with the structure of the hydrophobic tail. Among [G2] OGD regioisomer mixtures **3–5**, higher relative protein quantities were obtained when lipid-like tails were used. Our data indicate that the structural impact of OGD head groups and tails on protein purification can be extrapolated to alpha-helical and beta-barrel proteins. This leads us to the conclusion that tuning (i) the heterogeneity of the OGD batches and (ii) the structure of the OGD head and tail; are key parameters for optimizing the observable protein yields upon isolation under the experimental conditions employed.

**Preserving native lipid interactions**. While establishing the utility of OGDs for protein purification, we found that [G2] OGD regioisomer mixtures are less effective in isolating proteins from cell membranes than [G1] OGD regioisomer mixtures. This is in line with our hypothesis that increasing the size of the detergent head group decreases the detergent's ability to break lipid–lipid and protein–lipid interactions in membranes[4]. We therefore anticipate that tuning the structure of the OGD head and tail will not only control the obtainable protein yields, but also the ability to preserve protein interactions with native membrane lipids during isolation. These interactions can be probed using native MS since it enables individual protein–lipid binding events to be captured upon removal of detergents from the proteomicelle inside a mass spectrometer[25,29,30]. However, detergents that promote lipid co-purification, and maintain subunit interactions, are often difficult to remove from proteomicelles and vice versa[31]. As a consequence, detergents that enable both preservation of subunit interactions and easy detergent removal are not common.

With the aim of combining these two important features, we analyzed a series of bacterial membrane protein complexes that were extracted with the OGD batches **1–5** using a modified Q Exactive MS instrument[18]. The mass spectra obtained following the isolation of AqpZ with [G1] OGD batches revealed a well-resolved tetrameric complex in its *apo* form (Fig. 2c). Therefore, we conclude that the oligomeric state of AqpZ was retained during isolation. In the lower mass range of the spectrum AqpZ dimers of lower intensity were observed (Supplementary Fig. 6). This suggests that OGDs are also capable of solubilizing partially assembled states of oligomeric AqpZ. Such partial assemblies are commonly removed by using further purification techniques, such as size-exclusion chromatography (SEC)[32]. Mass spectra obtained from other bacterial membrane proteins, such as AmtB, MATE, OmpT, and OmpF, show exclusively the expected oligomeric states (Supplementary Figs. 2, 7–10, 14, and 15). In summary, our MS data highlight the utility of OGDs to preserve native oligomeric states of membrane proteins during purification.

Interestingly, poorly-resolved and broad charge state distributions were obtained for AqpZ upon extraction with individual [G1] OGD regioisomers **2a** and **2b** (Supplementary Fig. 7). Apparently, the [G1] OGD regioisomer mixture **2** (=**2a** + **2b**) is more suitable for the extraction and subsequent MS analysis of AqpZ than the individual [G1] OGD regioisomers **2a** and **2b**. As mentioned before in the case of AmtB, differences in extraction efficiency between **2**, **2a**, and **2b** were less pronounced. For all three OGD batches, mass spectra of comparable quality were obtained for AmtB (Supplementary Fig. 8). This demonstrates that the utility of OGDs for protein extraction is not necessarily limited to their regioisomer mixtures. If the targeted protein is sufficiently stable, individual OGD regioisomers can also be used for the purification and native MS analysis of membrane proteins. The ability to optimize the performance of OGDs for protein purification by changing the regioisomer ratios depends on the targeted protein.

From the [G2] OGD regioisomer mixture **3**, poor quality spectra and low yields were obtained, implying that the combination of a linear C18 alkyl chain and a [G2] head group is less suitable for protein isolation from cell membranes (Supplementary Fig. 9). In contrast, the combination of [G2] and lipid-like hydrophobic tails, e.g. **4** and **5**, gave rise to mass spectra assigned to lipid-bound states of tetrameric AqpZ complexes (Fig. 2c). The lipid masses agree well with those of cardiolipins (CDL) and phospholipids (PL) (Supplementary Table 2). These lipids were co-purified from cell membranes and are relevant for the structure and function of AqpZ[25,30]. We found similar trends in lipid preservation for AmtB and MATE. In contrast, MS spectra obtained from proteins that were purified with [G1] OGDs revealed a lower abundance of lipid-bound states (Fig. 2c, Supplementary Figs. 2, 10, 11). We conclude that tuning the structure of the OGD head group and tail enables control over the preservation of protein interactions with endogenous membrane lipids during protein isolation from cell membranes.

Furthermore, we investigated the stability of MATE-GFP and AqpZ-GFP against precipitation in MS buffer containing DDM, [G1] regioisomer mixture **1**, or [G2] OGD regioisomer mixture **4**. The stabilities of both proteins against precipitation in MS buffer were similar in all three detergent environments (Supplementary Fig. 12). Moreover, the isolated proteins were stable in solution and could be analyzed by native MS even after multiple freeze-thaw cycles. This further emphasizes the general utility of OGDs for the structural analysis of membrane proteins.

**OGD design and native MS.** Having established the utility of OGDs for protein purification and preservation of protein interactions with native membrane lipids during isolation, we evaluated their impact on the properties of native mass spectra. In contrast to the reference detergent DDM, resolved charge states were obtained for every membrane protein tested when OGDs were used during purification. This confirms that harsher activation conditions are needed to remove DDM from proteomicelles and shows that OGDs are removed more readily (Supplementary Fig. 13)[31].

To investigate charge-reducing properties of OGDs, we selected OmpF as model protein. We found that the average charge state ($z_{ave}$) of OmpF obtained upon MS analysis with [G1] OGD regioisomer mixture **1** (19+) is similar to that of $n$-octyl-ß-D-glucoside (19+), a detergent that is not associated with charge reduction (Fig. 3a, Supplementary Figs. 14 and 19)[31]. Further analysis of a mass spectrum obtained from OmpT upon purification with the [G1] OGD regioisomer mixture **1** supported this conclusion. The $z_{ave}$ of OmpT obtained after purification

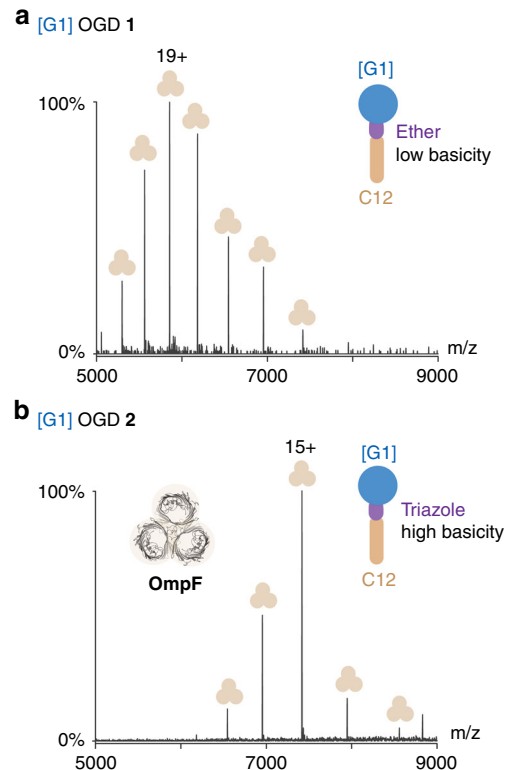

**Fig. 3 OGDs control membrane protein charge reduction. a, b** Mass spectra of trimeric OmpF were obtained upon detergent exchange from $n$-octyl-β-D-glucopyranoside to [G1] OGD regioisomer mixture **1** and [G1] OGD regioisomer mixture **2**. A substitution of the ether moiety in **1** by the more basic triazole in **2** leads to a substantial charge reduction of OmpF. Spectra were acquired using comparable instrument conditions (HCD energy: 200 V). Trimeric OmpT is indicated by beige symbols.

with **1** (11+) is similar to the most abundant protein charge state obtained from DDM (10+)[33], a detergent that is also not associated with charge reduction (Supplementary Fig. 15)[31].

In contrast, a significant reduction in $z_{ave}$ of OmpF from 19+ to 15+ is found when the ether linkage of **1** was substituted for a triazole unit in **2** (Fig. 3b). This charge reduction is expected because the ability of the linker to abstract charges (mainly protons) from protein ions is directly related to its basicity[23]. The $z_{ave}$ obtained from the [G1] OGD regioisomer mixture **2** (15+) was lower than the value obtained from the charge-reducing detergent C8E4 (17.5+, Supplementary Figs. 14 and 16). Further experiments on the individual [G1] OGD regioisomers **2a**, **2b**, and [G2] OGD regioisomer mixture **3** confirmed that protein charge reduction depends exclusively on the basicity of the OGD linker (Supplementary Fig. 14). Furthermore, the mass spectra obtained from AqpZ and AmtB upon isolation with the [G1] and [G2] OGD regioisomer mixtures **1–2** and **4–5** revealed that the ability to preserve lipid binding and protein charge reduction can be addressed individually, either by tuning the combination of head and tail or the choice of the linker, respectively (Supplementary Figs. 10 and 11).

In general, low activation MS conditions and reduced protein charge states are favored when performing native MS. Both parameters help to preserve native structural features of membrane protein complexes, because they decrease the effect of Coulomb-driven protein unfolding during detergent removal inside the mass spectrometer[20,34]. Members of individual detergent families, such as saccharide, polyethylene glycol, or amine N-oxide detergents, are either suitable for protein

purification, promote charge reduction, or preserve protein–lipid interactions during protein purification[31]. Individually, all of these properties are important, but no detergent to date embodies all of these desirable characteristics. We conclude that the OGDs present an example of a detergent family that unites the desirable features of saccharide, polyethylene glycol, and amine N-oxide detergents.

An important question prompted by this study is related to the increase in relative intensities of protein–lipid complexes detected by MS. Is this increase due to their stabilization in solution or in the gas phase? To answer this question we again focus on protein mass spectra obtained upon purification with [G1] and [G2] OGD regioisomer mixtures **1** and **4**. The spectra obtained for tetrameric AqpZ, for example, show similar protein charge states (Fig. 2c). Both detergents are not charge-reducing and the ionized protein-complexes experience similar collisional activation during detergent removal inside of the mass spectrometer[31]. Higher relative intensities of protein–lipid complexes were detected when the [G2] OGD regioisomer mixture **4** was used. The [G2] OGD regioisomer mixture **4** exhibits larger head groups than mixture **1**, as well as a more lipid-like tail (Fig. 2c). Furthermore, protein mass spectra obtained from [G1] and [G2] OGD regioisomer mixtures **2** and **5** confirm this observation. Both detergent batches reduce the charge of membrane proteins thus leading to lower energy collisional activation conditions within the HCD cell under the experimental conditions employed. However, protein–lipid complexes were again only obtained upon purification with [G2] OGD regioisomer mixture **5**, which exhibits larger head groups and a more lipid-like tail (Supplementary Fig. 11). This leads us to the conclusion that the increase in relative intensities of protein–lipid complexes depend more on their stabilization by the proteomicelle in solution rather than on charge effects in the gas phase.

Detergents exhibit individual delipidation properties and protein delipidation can also change with the time that is used to expose membrane protein–lipid complexes to detergents[35,36]. From our experience, detergents that co-purify substantial amounts of lipids are often not suitable for the straightforward MS analysis of membrane protein complexes. DDM, for example, is known to co-purify substantial amounts of lipids and requires harsh MS activation conditions to achieve sufficient detergent removal[35,37], which can hamper the detection of intact membrane protein complexes by MS[38]. In practice, the investigation of protein–lipid interaction by MS is addressed mainly in two ways: First, membrane proteins are delipidated step-wise with detergents that exhibit weak delipidating properties, such as DDM. To do so, protein–lipid complexes are repetitively purified by SEC, IMAC, or dialysis until mass spectra of sufficient quality are obtained[36,39]. This allows us to investigate membrane proteins in complex with co-purified membrane lipids[40]. In the second approach, membrane proteins are purified with detergents that exhibit strong delipidating properties, such as C8E4, OG, or LDAO[25,31]. If the targeted protein is sufficiently stable after delipidation, individual lipids can be added back to the sample solution[25,29]. Subsequently, MS analysis, gas-phase unfolding protocols, or functional assays allow us to study how the molecular structure of individual lipids affects the structure and function of membrane proteins[25,29,41,42].

In contrast to the above-mentioned methodologies, OGDs enable the straightforward analysis of interactions between membrane proteins and native membrane lipids directly after protein extraction and IMAC. Moreover, following our purification protocol, relative protein amounts and lipid binding interactions can be controlled practically by tuning the structure of the OGD head and tail (Fig. 2c, Supplementary Figs. 2, 10, and 11). This facilitates experimental access to either membrane proteins in complex with native membrane lipids or delipidated membrane proteins. In addition, OGDs can be readily removed from proteomicelles by collisional activation inside the mass spectrometer, which facilitates the MS analysis of membrane protein–lipid complexes in general. Finally, protein charge reduction can be tuned by varying the basicity of the linker between OGD head and tail. The ability to optimize the structure of OGDs for the purification and native MS analysis of protein–lipid interactions strengthens our anticipation that OGDs will facilitate the investigation of challenging membrane proteins in the future.

**Enabling step for GPCR research.** Having established the utility of OGDs for membrane protein research and native MS, we focus on one of the most challenging protein families in structural biology: GPCRs. A recent breakthrough came with the demonstration that native MS has the potential to elucidate how post-translational modifications and lipid interactions affect the structure and function of GPCRs[42,43]. However, research on GPCRs remains generally challenging, partly because of the limited range of detergents that facilitate their purification and analysis[44–46]. Addressing this challenge, we extracted NTSR1 fused with maltose-binding protein (MBP) and thioredoxin (TrxA) from *E. coli* membranes. Mammalian membrane proteins, such as GPCRs, are more prone to denaturation in the presence of detergents than bacterial membrane proteins[47]. During our analysis, DDM was therefore again used as a reference for comparison with our best performing [G1] and [G2] OGD regioisomer mixtures **1** and **4**.

Higher relative protein concentrations were obtained from the [G1] OGD regioisomer mixture **1**, which highlights its utility for extracting large GPCR quantities from biological membranes (Fig. 4). Native MS experiments revealed reduced relative intensities of protein–lipid complexes upon purification with the [G1] OGD regioisomer mixture **1** (Fig. 4). Enhanced relative intensities of lipid complexes were observed in mass spectra following purification with [G2] OGD regioisomer mixture **4**. Resolved mass spectra could not be obtained, however, following purification with DDM. This is in line with the results presented above for bacterial membrane proteins and underlines the utility of OGDs for tuning membrane protein and lipid co-purification under the experimental conditions employed.

Subsequent proteolytic removal of the MBP and TrxA tags led to extensive precipitation of NTSR1 in case of the [G2] OGD regioisomer mixture **4**. Only in the case of the [G1] OGD regioisomer mixture **1** and DDM it was possible to retain the solubility of NTSR1. To investigate the activity of NTSR1 in these detergents we used the chromophore-labeled agonist [5,6-FAM-NT(8-13)] and compared the results obtained in DDM and **1**. We found that agonist binding to NTSR1 was closely similar in the presence of **1** and DDM under the chosen experimental conditions (Supplementary Fig. 17). Further affinity MS experiments revealed that the chromophore-labeled agonist binds strongly to NTSR1, when the protein was purified with the [G1] OGD regioisomer mixture **1**. The $K_d$ value obtained from the NTSR1-agonist complex is in the nanomolar range and similar to data reported from cell-based assays (Fig. 4)[48]. Fluorescence polarization experiments supported this result (Supplementary Fig. 18). This underlines that the [G1] OGD regioisomer mixture **1** preserves functional characteristics of the receptor during purification[49]. Taken together, we conclude that the [G1] OGD regioisomer mixture **1** enables not only the straightforward purification and native MS analysis of NTSR1, but also the investigation of functional GPCRs. In light of these results, we anticipate that this OGD will serve as an enabling step for structural biology research on GPCRs.

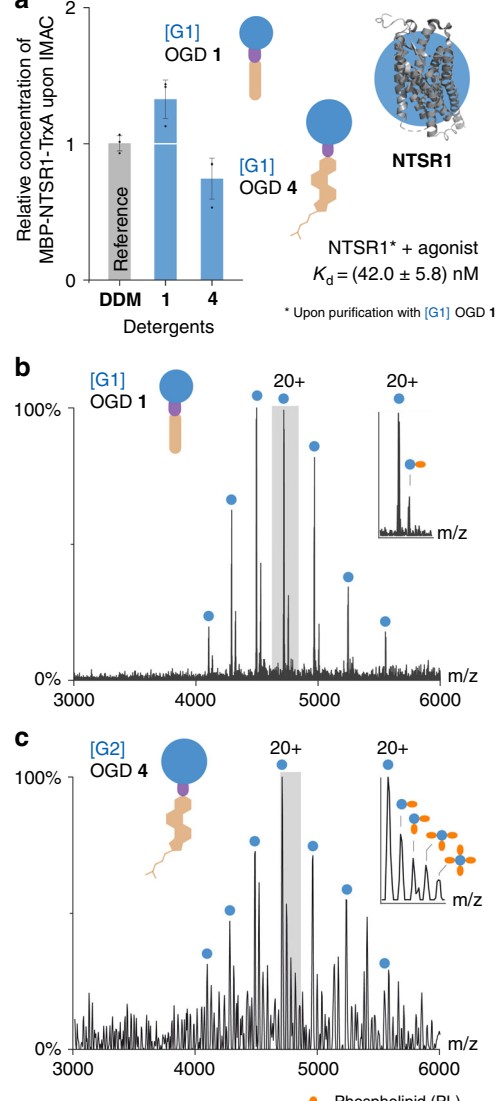

**Fig. 4 OGDs enable the purification of functional GPCRs. a** Higher relative concentrations of MBP-NTSR1-TrxA were obtained upon extraction and IMAC with the [G1] OGD regioisomer mixture **1**. The $K_d$ value of the complex is in the nanomolar range and similar to data reported from cell-based assays[48]. The [G1] OGD regioisomer mixture **1** preserves functional characteristics of the receptor during purification. **b**, **c** Native MS analysis of MBP-NTSR1-TrxA upon solubilization from cell membranes and IMAC purification with [G1] OGD regioisomer mixture **1** and [G2] regioisomer mixture OGD **4** show that the preservation of lipid binding can be controlled during purification by tuning the structure of the OGD head group and tail. Spectra were acquired using similar instrument conditions (HCD voltage: 200 V). MBP-NTSR1-TrxA is indicated by blue circles and phospholipids are indicated by orange circles. Data are shown with standard deviation (±s.d., $n = 3$). Source data are provided as a Source Data file.

## Discussion

In summary, we have introduced a library of OGDs to structural biology, which enables the isolation of a broad range of membrane proteins and facilitates their subsequent native MS analysis. Furthermore, these detergents serve not only as an enabling step for the purification and analysis of bacterial membrane proteins but also for the particularly challenging class of GPCRs. None of the OGDs investigated here is universally suitable for all applications in membrane protein research. However, OGDs present an example

of a detergent family whose molecular structures can be optimized for individual application in membrane protein research. This fine-tuning enables protein purification from membranes, preservation of lipid interactions during purification, refolding of membrane proteins, the facile MS analysis of membrane proteins and their lipid complexes, and confers protein charge reduction. Taken together, these attributes represent a significant step forward for the investigation of challenging membrane proteins and their interactions with native membrane lipids. Moreover, OGDs enable the purification of functional membrane proteins. In light of these findings, we anticipate that library-oriented detergent discovery of custom-made detergents for individual applications will be widely deployed in membrane protein research.

## Methods

**Synthesis and characterization of OGDs**. Detailed information on the synthesis of OGDs is given in the Supplementary Information (see Supplementary Methods). Critical aggregation concentration (cac) values of **1**–**5** were determined using dynamic light scattering (DLS)[50]. Serial dilutions with OGD concentrations between $10^{-8}$ and $10^{-2}$ M were prepared in MilliQ water. The samples were filtered (RC, 0.2 μm) and equilibrated for 16 h at room temperature prior to their analysis. The samples were transferred into a quartz cuvette (Quartz Suprasil, width × length: 2 mm × 10 mm) and analyzed with a Zetasizer Nano-ZS ZEN3600 (Malvern, UK). The instrument was operated with the Zetasizer Software (v7.11) and the following acquisition parameters were used: material (polystyrene latex), dispersant (water), sample viscosity parameters (use dispersant viscosity as sample viscosity), temperature (22.5 °C), equilibration time (120 s), cell type (quartz cuvettes), measurement angle (173° backscatter), measurement duration (manual), number of runs (11), run duration (10 s) number of measurements (3), delay between the measurements (0 s), data processing (general purpose, normal resolution). The derived count rate values obtained from three measurements (per concentration) were averaged and the logarithm of the derived count rate average was plotted against the logarithm of the OGD concentration. The double logarithmic plots showed two characteristic regions: (a) at flat region with low count rates at lower OGD concentrations and (b) a linear growth in the derived count rate at higher OGD concentrations. The individual regions were fitted to linear functions and the cac value was calculated from their intersection. The cac values are summarized in Supplementary Table 3.

**Membrane protein preparation for native MS**. Plasmids were provided by Idlir Liko, Hsin-Yung Yen, Kin-Kuan Hoi, and Jani Reddy Bolla. Plasmids were transformed into C43 (DE3) cells by mixing 1 μL of plasmid solution (100 ng/μL) with a 50 μL aliquot of C43 cells (purchased from Cambridge Bioscience). The mixture was incubated on ice for 30 min, heat shocked at 42 °C for 45 s, and cooled on ice for 2 min. LB Broth (450 μL of a 25 g/L aqueous solution) was added and the mixture was shaken with 180 rpm at 37 °C for 1 h. One 50 μL aliquot of this mixture was plated on an agar plate (agar medium composition: 25 g/L LB Broth and 15 g/L agar in water, supplemented with 100 μg/mL ampicillin) and the plate was stored over night at 37 °C. Up to five colonies were transferred into starter culture medium (5 mL of 25 g/L LB Broth, supplemented with 100 μg/mL ampicillin) and shaken with 180 rpm at 37 °C for 8 h. The starter culture was transferred into overnight culture medium (400 mL of 25 g/L LB Broth, supplemented with 100 μg/mL ampicillin) and the mixture was shaken with 180 rpm for 15 h at 37 °C. The overnight culture was transferred into in 12 L expression medium (12 × 1 L of 25 g/L LB Broth, supplemented with 100 μg/mL ampicillin)[25]. Cells were shaken with 180 rpm at 37 °C until an $OD_{600}$ between 0.7 and 1.0 was reached. Protein expression was induced by adding isopropyl-ß-D-thiogalactopyranoside (12 × 1 mL of a 0.5 M aqueous solution) and the cells were shaken with 180 rpm at 37 °C for four hours. Cells from a 12 L expression batch were harvested (5000 × $g$, 10 min), suspended in 100 mL buffer (20 mM Tris, 300 mM NaCl, 20% $v/v$ glycerol, pH = 7.4, protease inhibitor), and lysed using a Microfluidizer. After supernatant clarification (20000 × $g$, 20 min, 4 °C), the membranes were pelleted down (100000 × $g$, 2 h, 4 °C) and homogenized in 6 mL buffer B (20 mM Tris, 100 mM NaCl, 20% $v/v$ glycerol, pH = 7.4, protease inhibitor). Membrane aliquots (0.5 mL) were treated with buffer B (3.5 mL) and detergent stock solution (1 mL, c = 5w%, in water). Mixtures were agitated for 16 h at 4 °C. The supernatant was isolated by centrifugation (21000 × $g$, 40 min, 4 °C) and purified by IMAC. Empty spin columns (1.2 mL, Bio-Rad) were loaded with Ni-Agarose suspension (500 μL, Quiagen). The column was washed with water (3 × 500 μL), IMAC wash buffer (1 × 500 μL, 50 mM Tris, 200 mM NaCl, 20 mM imidazole, 10% $v/v$ glycerol, pH = 8), IMAC elution buffer (1 × 500 μL, 50 mM Tris, 100 mM NaCl, 500 mM imidazole, 10% $v/v$ glycerol, pH = 8), and IMAC wash buffer (5 × 500 μL). Columns were loaded with protein solutions, washed with IMAC wash buffer (5 × 500 μL) and IMAC buffer mixture (2 × 500 μL of wash/elution buffer, $v/v$, 9/1). Proteins were eluted with IMAC elution buffer (550 μL). Freshly eluted protein solutions were concentrated in Amicon Ultra 0.5 mL centrifugal filters, diluted with IMAC wash buffer (500 μL), and concentrated again.

Protein solutions were concentrated to equal volumes and relative protein concentrations were determined by UV/VIS spectroscopy using a microvolume photospectrometer (DeNovix, United Kingdom). Absorbance values were normalized to the values obtained from DDM and plotted against the detergent abbreviations ($A_{485}$ for AqpZ-GFP and MATE-GFP, $A_{280}$ for AmtB-MBP, BAM complex, and MBP-NTSR1-TrxA). Measurements were carried out in triplicate and the data were plotted with standard deviation (±s.d.). Corresponding data points were overlaid as dot plots in the bar charts. Protein purity was judged by SDS page analysis using MES buffer, unless otherwise indicated. MBP-NTSR1-TrxA was expressed in *E. coli* and purified as described before[42]. Buffer exchange into MS Buffer (200 mM NH4OAc, pH = 6.8) was achieved with 75 μL Zeba™ Spin Desalting columns (MWCO = 7 kDa, Thermo Fisher Scientific). All purification buffers were supplemented with detergents (Supplementary Table 4).

**Refolding of OmpT.** The plasmid was provided by Jani Reddy Bolla. The G236K/K237G mutant of OmpT was expressed in BL21 cells (DE3, purchased from New England Biolabs) in 12 L batches (12 × 1 L LB Broth, supplemented with 50 μg/mL kanamycin) as described above[27]. Cells were harvested (5000×g, 10 min), suspended in 30 mL suspension buffer (20 mM Tris, 150 mM NaCl, pH = 7.6, protease inhibitor), and lysed using a Microfluidizer. The insoluble material was spun down (5000 × g, 10 min, 4 °C), the supernatant was discarded, the pellet was suspended in 30 mL suspension buffer, and the insoluble material was spun down again. The supernatant was discarded and the insoluble inclusion bodies obtained from 12 L culture were suspended in buffer (50 mL of 20 mM Tris, 150 mM NaCl, 8 M urea, pH = 7.6, protease inhibitor) and stirred for 1 h at room temperature. The supernatant was clarified by centrifugation (20000 × g, 20 min). An aliquot of the so-obtained urea solution (1 mL) was mixed with refolding buffer (8 mL of 20 mM Tris, 150 mM NaCl, pH = 7.6, and 0.57 w% detergent) and agitated over night at 4 °C. The supernatant was clarified by centrifugation (4000 g, 30 min) and IMAC columns were prepared as described above using glycerol-free buffers. The columns were loaded with clarified supernatants and washed with IMAC wash buffer (2.5 mL of 50 mM Tris, 200 mM NaCl, 20 mM imidazole, pH = 8). OmpT was eluted with IMAC elution buffer (550 μL of 50 mM Tris, 200 mM NaCl, 250 mM imidazole, pH = 8). Relative protein quantities were determined by UV/VIS spectroscopy ($A_{280}$ for OmpT) and ratios between folded and unfolded OmpT were analyzed by SDS page analysis (Supplementary Fig. 4). Samples were not boiled prior analysis. For the enrichment of folded OmpT, the freshly eluted protein solutions were exchanged into IMAC wash buffer using a desalting column (column volume = 5 mL, GE Healthcare, product number: GE29-0486-84). So-obtained protein solutions were loaded onto freshly prepared IMAC columns and washed with IMAC wash buffer (1 mL). OmpT was eluted again using an intermediate imidazole concentration (1.5 mL of 50 mM Tris, 200 mM NaCl, 40 mM imidazole, pH = 8).

**CD spectroscopy.** Protein solutions obtained upon IMAC were transferred into CD spectroscopy buffer (100 mM NH4(HCO3), pH = 8) using desalting columns (column volume = 5 mL, GE Healthcare, product number: GE29-0486-84). Columns were washed with water (15 mL) and equilibrated with detergent-containing CD spectroscopy buffer (10 mL, 2xcac). Protein solutions obtained upon IMAC (~ 0.5 mL) were injected manually into the columns using syringes. The proteins were eluted with CD spectroscopy buffer (10 mL) and fractions were collected (fraction size = 1 mL). Protein-containing fractions were identified by UV spectroscopy, combined, and concentrated to a final protein concentration of 5–10 μM. So-obtained protein solutions were loaded into cuvettes (Quartz Suprasil, volume = 300 μL, layer thickness = 1 mm). The CD spectrometer (Chirascan, USA) was purged with nitrogen overnight and turned on 30 min before use together with the sample cooler. The following experimental parameters were used: temperature (22.5 °C), wavelength range (200–260 nm), step size (0.5–1 nm), scan speed (0.5 s/point), bandwidth (1 nm), and repeats per sample (4). The average CD intensity of four scans was plotted against the wavelength. Detergent-containing CD spectroscopy buffers were used as blanks. Data were acquired with Pro-Data Chirascan V4.5 and analyzed with Origin V9.1. For the comparison of batches with different protein concentrations, the CD intensity values were converted into mean residue ellipticity values as described elsewhere (Supplementary Fig. 3)[51]. If necessary, the buffer exchange step was repeated to reduce remaining imidazole and salt contaminants.

**Monitoring the activity of OmpT.** The activity of OmpT was assessed by monitoring the time-dependent cleavage of a self-quenching fluorescent peptide Abz-ARRAY-Tyr(NO2)-NH2 (Biomatik, custom synthesis) in which "Abz" abbreviates *o*-aminobenzoyl and "Tyr(NO2)" abbreviates 3-nitrotyrosine[27]. The following components were mixed in chambers of a 96 well plate (Greiner 96F-Bottom): assay buffer (233.5 μL of 10 mM Bis-Tris, 5 mM EDTA, pH = 6.5), OmpT (10 μL of a 10 μM OmpT solution in 100 mM (NH4)HCO3, pH = 8), LPS (10 μL of a 5 mg/mL solution in H2O), and Abz-ARRAY-Tyr(NO2)-NH2 (9 μL of a 980 μM aqueous solution). For LPS-free assay conditions, a larger assay buffer volume was used (243.5 μL). Assay buffers and protein solutions were supplemented with detergent (2xcac). The peptide solution was added last and all ingredients were mixed before analysis resulting in a dead time of ~25 s. Time-dependent cleavage of the peptide was monitored with a CLARIOstar microplate reader (BMG Labtech). The following experimental parameters were used: bottom optic, focal height (4.1 mm),

excitation wavelength (325 nm), emission wavelength (450 nm), numer of cycles (70), cycle duration (10 s), temperature (25 °C), No. of flashed per well (20), gain (1000), and settling time (0.5 s). Data were acquired with CLARIOstar® V5.4 and analyzed with MARS V3.3 and Origin V9.1.

**Charge reduction experiments.** OmpF was purified in *n*-octyl-β-D-glucopyrano-side[52]. Detergent exchange to **1–3** was performed on a Superdex 200 10/300 GL column (product number: 17-5175-01) in MS buffer (200 mM NH4OAc, pH = 6.8) supplemented with detergent (Supplementary Table 4). Eluted proteins were concentrated using Amicon Ultra 0.5 mL centrifugal filters.

**Mass spectrometry analysis.** Membrane proteins were analyzed under comparable instrumental conditions using a modified Q Exactive instrument[18]. Instrumental parameters were as follows: injection flatapole (7.9 V), inter flatapole lens (6.9 V), bent flatapole (5.9 V), transfer multipole (4 V), capillary voltage (1.2 kV), source temperature (100–250 °C), voltage applied to the C-trap entrance lens (5.8 V), higher-energy collisional dissociation (HCD) cell voltage (200 V), HCD cell pressure ($9 \times 10^{-10}$ mBar), noise level parameter (3), microscans (1–10), and resolution (17500). UniDec was used for background substraction and smoothing of recorded spectra[53]. Theoretical and experimental masses of detected species are summarized in Supplementary Table 5. Data analsis was performed with Xcalibur V2.2 and Origin V9.1.

**$K_d$ measurements.** NTSR1 purified in [G1] OGD regioisomer mixture **1** was buffer exchanged into binding buffer (10 mM Hepes, 150 mM NaCl, 2xcac of [G1] OGD regioisomer mixture **1**, pH = 8) and incubated on ice with different concentrations of [5,6-FAM-NT(8-13)] (Caslo ApS, custom synthesis, peptide sequence: (5,6-FAM)-RRPYIL) ranging from 2 nM to 250 nM. The unbound peptide was separated using 75 μL Zeba™ Spin Desalting columns (MWCO = 7 kDa, Thermo Fisher Scientific) and the protein bound to the labeled peptide was collected. The amount of peptide was determined using fluorescence polarization and affinity MS. Fluorescence measurements were made using a PHERAstar FSX (BMG Labtech) microplate reader and the data were analyzed using PHERAstar FSX Mars data analysis software. For affinity MS, a system composed of a Vanquish™ Flex Quaternary UHPLC quaternary pump, a Vanquish™ Split Sampler, and a Q Exactive™ Hybrid Quadrupole-Orbitrap mass spectrometer was used. An Accucore™ biphenyl reversed-phase analytical column (2.1 × 100 mm, 2.6 μm) was used as stationary phase. The flow rate was set to 200 μL/min and the column temperature was set to 40 °C. Buffer A (water with 0.1% formic acid) and buffer B (methanol with 0.2% acetic acid) were used as mobile phases. The gradient was programmed as follows: 0–0.15 min, 10% B; 0.15–3.5 min, 10–95% B; 3.5–4.25 min, 95% B and 4.25–4.3 min, 10% B with a final run time of five minutes. The Q Exactive mass spectrometer was operated in positive mode and the ionization conditions were as follows: capillary temperature (250 °C), vaporizer temperature (250 °C), spray voltage (3.5 kV), auxiliary gas pressure (10 arbitrary units) and sheath gas pressure (45 arbitrary units). Peptides were detected using data-dependent acquisition mode in which the five most intense precursor ions from a full MS scan were selected for fragmentation by collision induced dissociation (CID). Full MS scans were performed with a resolution of 70,000 at 200 *m/z*. The *m/z* range was set to 400–1300. The normalized CID collision energy was 35% for a doubly charged precursor ion. The isolation width was set to 2 *m/z* and the activation time to 10 ms. MS data were analyzed using TraceFinder™. MS and fluorescence intensities were normalized and plotted against the concentration. The data were fitted to a one-site binding model in order to obtain $K_d$ values (Supplementary Fig. 18).

**Reporting summary.** Further information on research design is available in the Nature Research Reporting Summary linked to this article.

## Data availability
Data supporting the findings of this manuscript are available from the corresponding authors upon reasonable request. A reporting summary for this Article is available as a Supplementary Information file. Source data for Figs. 2b and 4, Supplementary Figs. 1b, 2a, 3, 4, 5a, 12, 17, 18, and Supplementary Tables 1–5 are provided with the paper as a Source Data file. Mass spectrometry raw data are available on the OSF website (project name: Modular Detergents Tailor the Purification and Structural Analysis of Membrane Proteins Including G-protein Coupled Receptors; https://doi.org/10.17605/OSF.IO/TJXSR).

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

## Acknowledgements

Financial support was provided by the Focus Area Nanoscale of the Freie Universität Berlin, Fonds der Chemischen Industrie (FCI), and ERC Advanced Grant No. 695511 (ENABLE). The authors acknowledge K. Goltsche, M. Selent, C. Fasting, M. Grabarics, P. Winchester, D. Wu, M. Agasid, V. Wycisk, A. Hardy, D. Saman, and the Core Facility BioSupraMol of the Freie Universität Berlin for continuous support. N. G. Housden and C. Kleanthous are acknowledged for providing OmpF. J.G. is supported by a Junior Research Fellowship at The Queen's College (Oxford). R. A. Drywood, A. Bowen, Mathys & Squire, Oxford University Innovation, A. Hübner, A. Schoberth, and the Patent and License Service of the Freie Universität Berlin are gratefully acknowledged for continuous support during the patent application process.

## Author contributions

L.H.U, S.E., R.H., and K.P. designed the detergents; L.H.U. synthesized and characterized the detergents; L.H.U., I.L., H.-Y.Y., J.R.B., C.V.R., and K.P. conceived the experiments;

L.H.U and I.L. performed the experiments; H.-Y.Y., K.-K.H., J.G., M.-P.S., F.G.A., J.R.B., and D.S. supported the experiments; all authors co-wrote the paper; L.H.U. and I.L. contributed equally to this work.

## Competing interests

The authors declare no competing interests. The Freie Universität Berlin and University of Oxford filed a joint patent application related to the application of OGDs for the purification and structural analysis of membrane proteins and soluble proteins (GB1814356). Carol Robinson and Joseph Gault provide consultancy services for OMass Therapeutics and Idlir Liko and Hsin-Yung Yen are employees of that company.
