## [Peer Review File · Nature Communications]

Reviewers' Comments:

Reviewer #1:

Remarks to the Author:

The manuscript reports a novel class of detergent, oligoglycero detergents (OGDs), that can be used to extract and purify membrane proteins. By altering the building blocks including the hydrophilic head, the linker and the hydrophobic tail, authors were able to "fine-tune" the properties of the resulting detergent and applied these detergents in five different proteins. The small library authors created covers different kinds of building blocks, which hint the underlying reasonings that governing the behavior of these detergents and may be helpful in further development of OGDs. All experiments were performed in triplicates, which is sufficient in such studies. Generally, the topic is of broad interest, not only to the field of native mass spectrometry, but also to the field of membrane protein extraction and purification. The manuscript fits in the scope of Nature Communications, and can be considered for publication after addressing these concerns listed below.

1. Authors used a set of numbering abbreviations of 1 - 5 to represent their synthesized detergents. These abbreviations were not well defined in the manuscript but defined systematically in Supplementary Table 1 and 3. It will be very helpful for the authors to define these structures in the manuscript, maybe as part of Figure 1.

2. At the end of Page 5, authors referred to Figure 2a in Supplementary Information. To my understanding, the information that authors tried to refer to is in Supplementary Figure 1. Please double-check.

3. In Page 6, authors compared the extraction efficiency between 2 and 2a & 2b. It is clear that relative protein quantities of detergent 2 is greater than that of 2a & 2b for AqpZ-GFP in Figure 2b. But such tendency is not clear for AmtB-MBP (SI, Figure 1a). Is there a reason for that? It is risky to draw the conclusion that stereo-heterogeneity is the key unless the authors can properly comment on that. Moreover, authors did not show the results for MATE in the SI. It will be helpful to compare the results for all three proteins together.

4. The authors state on page 6 line 11 "We speculate that the OGD regioisomer mixtures resemble the heterogeneity of biological membranes more closely, which improves protein extraction..." This statement is vague and requires clarification, best by example, to show lipid heterogeneity in a membrane and how that is mimicked by OGDs.

5. Also on page 6 line 15, the authors state: "Taken together, we conclude that tuning i) the heterogeneity of the OGD batches and ii) the structure of the OGD head and tail; are key parameters for optimizing the observable protein yields upon isolation." This is a premature conclusion. It is only based on a comparison of the OGDs and DDM.

6. The above points are examples of exaggerated claims and of statements needing justification. In the introduction: "This makes it increasingly difficult to define design guidelines and leads to the universal dogma that selection of detergents depends more upon empirical factors rather than on scientific principles. To address this shortcoming, we introduce the family of oligoglycerol..." And in the conclusion: "the OGDs reported here present the first example of a detergent family whose optimization for membrane protein research depends more on scientific principles rather than on empirical factors." I fail to see the justification for these claims. What scientific principles are invoked to allow optimization: vapor pressure, critical micelle concentration, hydrogen bonding, dipole moment, solubility...? There is limited discussion of "scientific principles" that lead to predictive capability. Furthermore, the authors tout the modular nature of the OGDs, but aren't most lipids and detergents modular?

7. In Page 6 line 19, authors refer to Supplementary Figure 1 for the isolation of AmtB and MATE.

However, Figure 1 in SI covers the AqpZ-GFP and AmtB-MBP, but not MATE. Authors should add the corresponding data for MATE into SI Figure 1.

8. In the end of Page 6, authors mentioned that [G2] OGDs are less effective in protein isolation than [G1] OGDs. In Supplementary Figure 2, however, the results contradict with the conclusion above (see SI Figure 2 Caption). Is it because MATE is smaller than the other two proteins? It will be helpful if the authors can comment on this point.

9. In Figure 2b, authors use cartoons to represent the two classes of detergents with different head groups ([G1] and [G2]). The [G2] OGDs cartoon has two chains in its hydrophobic tails, which is not the proper illustration for 3 and 4 since both of these two compounds comes with single-chained tail. Authors should consider redesigning the cartoon since it can be misleading.

10. In Page 8 line 9-13, authors mentioned that regioisomer mixture 2 causes less perturbation as compared with 2a and 2b. SI Figure 4 shows the corresponding spectra for 2a and 2b, but not 2. Authors should provide the spectrum using regioisomer mixture 2 to support their statement. Moreover, it will be helpful to show the spectra for the other two proteins (AqpZ-GFP and MATE) to support and generalize the statement. It will be interesting to compare the spectra for AmtB-MBP, where the difference between 2 and 2a, 2b is not as significant as that of AqpZ-GFP.

11. In Page 9 line 7-10, authors mentioned that harsher activation is needed for removal of DDM as compared with OGDs. Authors refer to SI Figure 4 - 6 to support their statement. To my understanding, authors should refer to SI Figure 3 - 6 and specify the Collision voltages for the spectra in SI Figure 3 - 5 to support their statement.

12. In Page 9 line 13, authors mentioned the average charge state of OmpF is +18. It should be +19 from SI Figure 7. Similar conclusion on charge reduction can be reached from SI Figure 3a, which can be added to the discussion.

13. In Figure 4a, it will be helpful for the authors to show the result of DDM for comparison.

There are also some minor points that need attention.

1. At the beginning of Page 4, authors should provide full name for "NTSR1" since it has not been defined before.

2. At the end of Page 5, IMAC should stand for immobilized metal (ion) affinity chromatography. Authors missed "affinity".

3. In SI Figure 1, there are ten columns in the gel picture, for which all the columns are properly labeled except the seventh. The 7th column was labeled as "-". Please specify/explain the label in the Figure caption.

4. In the last page, initials for author "Hsin-Yung Yen" is spelled incorrectly as "H.-Y.J." in "Author Contributions".

Reviewer #2:

Remarks to the Author:

The paper by Urner et al addresses an important issue in membrane protein structural characterization. Detergent conditions need to be optimized on a per protein level which can be both a time consuming and seemingly random process at times given the variety of different classes of detergent. The application of a modular, tunable, detergent class for structural biology studies is of great interest. I think this manuscript is well suited to nature communications and its

broad audience however a few additional controls should be included, as detailed below.

The authors compare the total protein extraction of the OGD detergents to the conventional detergent DDM, and then go on to compare the amount of endogenous lipid retained between the best performing G1 and G2 OGD detergents. However, the authors present seemingly inconsistent information for DDM; "In contrast to the reference detergent DDM, resolved charge states were obtained for every membrane protein tested when using OGDs (Supplementary Fig. 4 – 6)." In SI Figure 6 an unresolved MS is shown. However, in SI Table 1 the authors give an average charge state for Aqpz-GFP and the mass of lipids binding in DDM. This suggests they must have been able to resolve the MS under some conditions. The authors should provide this MS. This must be included so that a full comparison between the different detergents can be made.

The authors state that the native state of the protein is retained in these detergents. However, in Figure 2c only the m/z range containing the tetramer is shown. The m/z range should include the low mass range as this is needed to show that only the native oligomeric state is produced. The comparison to DDM mentioned above should also include the extended m/z range.

For the charge reduction experiments the authors should include a comparison with the known charge reducing detergent C8E4.

For all detergents except OGD1, 2x the critical aggregation concentration is used. For OGD1 only 1x the CAC was used. The authors should add an explanation for this into the SI.

For NTSR1, the authors only show extraction with OGDs and no comparison with DDM. However, they then go on to compare the binding activity of NTSR1 in DDM and OGD1. The authors should therefore also include the extraction efficiency in DDM. They also do not show any MS comparison with DDM. This should be included.

Minor comments:

Abstract- "even though a large variety of detergents has been developed" should be "even though a large variety of detergents have been developed"

Page 4 line 1-define NTSR1 at its first use.

Page 5-Given the broad readership of this journal I think it is essential to explain why tuning the charge states of membrane proteins in the MS is of great interest to the native MS community. In addition, describing what "favorable characteristics for native MS" are is essential. What metrics are the authors using here to discern which detergents are the most favorable.

Page 5- "Different constructs of the aquaporin channel (AqpZ), the ammonia channel (AmtB), and the multidrug efflux pump (MATE) were extracted from E. coli membranes using 1 – 5 (Figure 2a, Supplementary Information)" Should this sentence be referencing the SI? If so what in the SI?

The mass range should be increased in all SI figures to better demonstrate that only the native oligomer is observed and to also show any detergent interference. In addition, I would suggest the authors keep their symbol size consistent in the figures- particularly SI Figure 3 as it stands currently the smaller symbols can be confused for monomers- as they appear to be one small pink triangle as opposed to the grey and pink larger symbol.

The authors should also label other high intensity peaks in the MS-e.g. in SI Figure 5 b bottom panel-please label the peaks between 6000-7000 m/z

In the methods can the authors clarify if the same HCD energy was used for every MS acquired, if so state it in methods. If not it should be added to figure captions.

Reviewer #3:

Remarks to the Author:

In the publication "Modular Detergents Tailor the Investigation of Membrane Proteins Including G-protein Coupled Receptors", Urner et al. describe the synthesis of several oligoglycerol detergents (OGDs) and their utility for solubilising membrane proteins for purification and subsequent analysis by native mass spectrometry. The authors demonstrate the utility of these detergents for the solubilisation/mass spectrometric analysis of several integral membrane proteins, including the analysis of endogenous lipids carried through purification protocols. I have a number of concerns about the manuscript, including the level of interest of this work to the broad readership of Nature Communications, and suggest that revisions need to be made prior to publication to strengthen the evidence for the conclusions made. The work presented in this manuscript does not present a 'silver bullet' solution to the problem of detergent screening for native MS of membrane proteins, and whilst it is exciting that the authors have identified a new class of detergents capable of solubilising bacterial membrane proteins, there remains some gaps in the characterisation of their properties.

Major Comments

- The title of the manuscript is misleading and should be changed. What do the authors mean by 'Tailor the Investigation'? The title suggests that the OGDs have been validated for use by many structural techniques, but the authors mostly present native mass spectrometry data (with some other limited solution based data). The title should reflect this. I also don't see why the authors specifically mention G-protein coupled receptors in the title.
- For AqpZ, AmtB and MATE purifications, membranes were solubilised with OGDs for 16 hours. Is there a reason why the solubilisation was so long? Have the authors tested the solubilisation efficiency as a function of time? Is this comparable with conventional detergents?
- It is very confusing when the authors refer to "two OGD regioisomers" and a "mixture of detergent regioisomers". This must be clarified throughout the manuscript. Perhaps Figure 1 could be improved in a way that more clearly depicts the structure and heterogeneity of the detergent mixtures used.
- OGDs in no way resemble the heterogeneity and complexity of the biological membrane – references to this should be removed.
- In Supplementary Figure 1, purification data for MATE are not shown.
- In Supplementary Figure 2, did the authors identify the lipids bound to MATE?
- Are the in-solution stoichiometries identical for the proteins purified in each detergent? Are the samples homogeneous (e.g. using SEC-MALS), are their structures identical (e.g. using CD), and are the purified proteins (more) stable in the OGDs? How do these parameters compare with conventional detergents? In Supplementary Figure 3, for example, varying amounts of trimeric and monomeric AmtB are detected by native MS, do these monomers result from gas-phase dissociation or are they also present in solution?
- On page 8 the authors state that poorly-resolved charge state distributions were observed for AqpZ upon extraction with individual regioisomers of 2a and 2b and attribute this to perturbation of the protein structure during isolation. Do the authors have any evidence for this? Couldn't this also be attributed to the individual regioisomers requiring additional activation to release them from the protein?
- Could the OGDs be used to solubilise OmpF directly from membranes? All other tested IMPs were α -helical, and demonstrating the ability of OGDs to solubilise β -barrel proteins would further emphasise their generic utility.
- Does charge reduction alter the apparent affinities of the protein-ligand interactions detected by MS? The authors have previously implemented an OmpF-peptide binding experiment (e.g. Housden et al., Science, 2013). Are the measured affinities in the OGDs different than OG, and do they differ between detergents? The same question could be asked for apparent affinities measured for lipids titrated into the OGD-solubilised IMPs. Is the increased lipid observed in some OGDs due to the protein-lipid complexes being stabilised in different detergents in the gas phase (or in solution too)? Have the authors used any other methods to quantify the lipids extracted with the membrane proteins, apart from native MS?

- What yield is obtained in DDM relative to OGDs for NTSR1? The comparison presented in Figure 4a doesn't appear to be relative to DDM extraction, and in Supplementary Table 1 data for only detergent 1 is shown, whereas detergent 4 was also used to solubilise NTSR1 (Fig 4a), and DDM purified protein was used in binding assays as a comparison (Supplementary Figure 8). Additionally, were multiple biological replicates performed as error bars are not shown like in other figures?
- On page 11, the authors state "the finding that high relative intensities of lipid complexes were obtained upon isolation with 1 is not surprising, since it is known that lowering the extraction time effectively lowers protein delipidation". If that is the case, do lower extraction times with G1 OGDs for the other proteins studied result in increased lipid being retained throughout purifications? If G2 OGDs are less effective at solubilising membranes then I presume the effective extraction time is reduced (as it takes longer to solubilise proteins from the bilayer) so this could explain why the authors see increased amounts of bound lipid with these detergents.
- The data presented for agonist binding is weak. Could the authors use their fluorescein labelled peptides to determine K_d values, for example by fluorescence polarisation? Is the affinity perturbed in OGDs?
- Could the authors comment on the ratio of regioisomers chosen in Supplementary Table 2? Could these be tuned to give more favourable properties? Were these ratios selected in any particular way via the synthetic route?
- In Supplementary Table 3, why was only 1xcac used for ODG 1, whereas 2x cac amounts were used for the other detergents?
- Did the authors employ smoothing and background subtraction during data analysis? In some instances signal-to-noise is poor, could the authors comment on why this is the case?
- The manuscript doesn't really have a discussion section. The authors should think about putting their work a bit in context with the literature more clearly. What sort of mass spectrometry experiments might it be useful to try retain lipid binding events? Why is it beneficial to not have to perform detergent exchange? What other detergents or purification systems for membrane proteins have been investigated, why are OGDs better? What other applications might benefit from custom detergents, and how?

Minor Comments

- Bottom line of p3 – Remove the word "Remarkably".
- Four lines from bottom of p4 – Replace the word "outstanding"
- Page 5 – Why is it of interest to tune the charge states of membrane proteins in mass spectra? Could the authors expand on this statement?
- Page 15 – Change "comparative instrumental conditions" to "comparable instrumental conditions"

In detail, we respond to the reviewers comments as follows:

Major Comments of Reviewer #1:

The manuscript reports a novel class of detergent, oligoglycero detergents (OGDs), that can be used to extract and purify membrane proteins. By altering the building blocks including the hydrophilic head, the linker and the hydrophobic tail, authors were able to “fine-tune” the properties of the resulting detergent and applied these detergents in five different proteins. The small library authors created covers different kinds of building blocks, which hint the underlying reasonings that governing the behavior of these detergents and may be helpful in further development of OGDs. All experiments were performed in triplicates, which is sufficient in such studies. Generally, the topic is of broad interest, not only to the field of native mass spectrometry, but also to the field of membrane protein extraction and purification. The manuscript fits in the scope of Nature Communications, and can be considered for publication after addressing these concerns listed below.

1. Authors used a set of numbering abbreviations of 1 - 5 to represent their synthesized detergents. These abbreviations were not well defined in the manuscript but defined systematically in Supplementary Table 1 and 3. It will be very helpful for the authors to define these structures in the manuscript, maybe as part of Figure 1.

We thank the referee for this suggestion and added systematic descriptions of our detergents to Figure 1.

2. At the end of Page 5, authors referred to Figure 2a in supplementary information. To my understanding, the information that authors tried to refer to is in Supplementary Figure 1. Please double-check.

We thank the referee for raising this point and corrected the figure numbering within the revised manuscript.

3. In Page 6, authors compared the extraction efficiency between 2 and 2a & 2b. It is clear that relative protein quantities of detergent 2 is greater than that of 2a & 2b for AqpZ-GFP in Figure 2b. But such tendency is not clear for AmtB-MBP (SI, Figure 1a). Is there a reason for that? It is risky to draw the conclusion that stereo-heterogeneity is the key unless the authors can properly comment on that. Moreover, authors did not show the results for MATE in the SI. It will be helpful to compare the results for all three proteins together.

Following the referee’s concern, we performed a P-test to validate the statistical significance of our data. Based on the Analysis of Variance (ANOVA) test, the following P-values were obtained for relevant data set combinations:

P-value of **2** vs **2a** = 0.002

P-value of **2** vs **2b** = 0.0005

P-value of **2a** vs **2b** = 0.004

All P-values are lower than 0.05 and differences in relative protein amounts obtained upon extraction and IMAC can therefore be considered as significant. However, the differences in relative protein amounts obtained for AmtB are indeed less pronounced than for AqpZ. We addressed this point in the revised manuscript as follows: “We found similar trends in case of AmtB (Supplementary Fig. 1). However, differences in extraction efficiency between the regioisomer mixture **2** and the

individual isomers **2a** and **2b** were less pronounced than in the case of AqpZ, but still statistically significant (Supplementary Table 1).”

The P-test data were added to the supplementary information (see Supplementary Table 1) and Source Data file. #

4. The authors state on page 6 line 11 “We speculate that the OGD regioisomer mixtures resemble the heterogeneity of biological membranes more closely, which improves protein extraction....” This statement is vague and requires clarification, best by example, to show lipid heterogeneity in a membrane and how that is mimicked by OGDs.

In our opinion, the suggestion to show lipid heterogeneity in a membrane and how that is mimicked by OGDs is difficult to implement in the Figure. We therefore prefer to remove our speculation in the revised version of the manuscript.

5. Also on page 6 line 15, the authors state: “Taken together, we conclude that tuning i) the heterogeneity of the OGD batches and ii) the structure of the OGD head and tail; are key parameters for optimizing the observable protein yields upon isolation.” This is a premature conclusion. It is only based on a comparison of the OGDs and DDM.

To explain more clearly how we come to these conclusions, we extended our discussion in the revised version of the manuscript as follows: “Interestingly, protein quantities obtained from **2**, which contained two OGD regioisomers, were higher than that of the individual regioisomers **2a** and **2b**. We found similar trends in case of AmtB (Supplementary Fig. 1). However, differences in extraction efficiency between the regioisomer mixture **2** and the individual isomers **2a** and **2b** were less pronounced than in the case of AqpZ, but still statistically significant (Supplementary Table 1). To the best of our knowledge, such an increase in solubilized protein quantities arising from a mixture of detergent regioisomers has not been reported previously. [...] Taken together, OGD batches containing smaller head groups, such as [G1] OGD regioisomer mixtures **1 – 2**, led to higher protein quantities than OGDs batches with larger head groups, such as [G2] OGD regioisomer mixtures **3 – 5**. Furthermore, protein quantities obtained among [G2] OGD batches vary with the structure of the hydrophobic tail. Among [G2] OGD regioisomer mixtures **3 – 5**, higher relative protein quantities were obtained when lipid-like tails were used. Our data indicate that the structural impact of OGD head groups and tails on protein purification can be extrapolated to alpha-helical and beta-barrel proteins. This leads us to the conclusion that tuning i) the heterogeneity of the OGD batches and ii) the structure of the OGD head and tail; are key parameters for optimizing the observable protein yields upon isolation under the experimental conditions employed.”

To outline the role of DDM in our analysis we changed the relevant sentence in the manuscript as follows: “The relative protein amounts were determined by UV/VIS spectroscopy. Subsequently, the relative protein amounts obtained from **1 – 5** were compared with *n*-dodecyl- β -D-maltoside (DDM). This detergent enables the purification of high protein yields and is a current standard in structural biology (Fig. 2b)²⁶.” #

6a. The above points are examples of exaggerated claims and of statements needing justification. In the introduction: “This makes it increasingly difficult to define design guidelines and leads to the universal dogma that selection of detergents depends more upon empirical factors rather than on scientific principles. To address this shortcoming, we introduce the family of oligoglycerol....” And in the conclusion: “the OGDs reported here present the first example of a detergent family whose optimization for membrane protein research depends more on scientific principles rather than on empirical factors.” I fail to see the justification for these claims.

We thank the reviewers for raising this point and justify our claims as follows:

First, we outline why it is increasingly difficult to define design guidelines for detergents in membrane protein research by expanding the introduction as follows: “The number of detergent families is staggeringly diverse and their chemical variety makes it increasingly difficult to define design guidelines to predict the utility of detergents for purifying membrane proteins. Furthermore, the concentration of detergent used in purification protocols is often adjusted to a multiple of its’ *critical aggregation concentration (cac)*⁹. Interestingly, however, detergents with similar *cac* values can exhibit opposing compatibilities with proteins. For example, sodium dodecyl sulfate (SDS) and tetraethylene glycol mono-octyl ether (C8E4) have similar *cac* values. However, SDS is a strong protein denaturant while C8E4 preserves native structural features of various membrane proteins. This renders an estimation of the detergents’ utility for protein purification according to their *cac* values difficult. Together these attributes contribute to the emergence of the universal dogma that the selection of detergents depends more on empirical factors than on scientific principles^{10, 11}.”

To further underline why OGDs present the first example of a detergent family whose optimization for membrane protein research depends more on scientific principles, we specified the conclusion of our revised manuscript as follows: “None of the OGDs investigated here is universally suitable for all applications in membrane protein research. However, OGDs present the first example of a detergent family whose molecular structures can be optimized for individual application in membrane protein research. This fine-tuning enables protein purification, preservation of lipid interactions during purification, facilitates MS analysis of membrane proteins and their lipid complexes, and confers protein charge reduction.” #

6b. What scientific principles are invoked to allow optimization: vapor pressure, critical micelle concentration, hydrogen bonding, dipole moment, solubility...? There is limited discussion of “scientific principles” that lead to predictive capability.

For further explanation of scientific principles that allow us to tune OGDs for the above-mentioned applications see reply to Reviewer #2, No. 3b.

7. In Page 6 line 19, authors refer to Supplementary Figure 1 for the isolation of AmtB and MATE. However, Figure 1 in SI covers the AqpZ-GFP and AmtB-MBP, but not MATE. Authors should add the corresponding data for MATE into SI Figure 1.

We added requested purification data to the revised Supplementary Fig. 2 and the Source Data file was updated accordingly.

8. In the end of Page 6, authors mentioned that [G2] OGDs are less effective in protein isolation than [G1] OGDs. In Supplementary Figure 2, however, the results contradict with the conclusion above (see SI Figure 2 Caption). Is it because MATE is smaller than the other two proteins? It will be helpful if the authors can comment on this point.

We comment on this point in the revised manuscript as follows: “[...] the relative protein concentrations of AqpZ obtained from [G2] OGD regioisomer mixtures **3 – 5** were lower than those obtained from [G1] OGD regioisomer mixtures **1 – 2** or DDM. Similar results were obtained for the isolation of AmtB and MATE (Supplementary Fig. 1-2).”

9. In Figure 2b, authors use cartoons to represent the two classes of detergents with different head groups ([G1] and [G2]). The [G2] OGDs cartoon has two chains in its hydrophobic tails, which is not the proper illustration for **3** and **4** since both of these two compounds comes with single-chained tail. Authors should consider redesigning the cartoon since it can be misleading.

Structural differences among [G2] OGD batches **3 – 5** are as follows: [G2] OGD **3** contains a linear C18 hydrocarbon chain, [G2] OGD **4** contains a cholesterol moiety, and [G2] OGD **5** contains two C12 hydrocarbon chains. To visualize these differences, we redesigned the cartoons of [G2] OGDs **3 – 5** in Fig. 2 accordingly.

10a. In Page 8 line 9-13, authors mentioned that regioisomer mixture **2** causes less perturbation as compared with **2a** and **2b**. SI Figure 4 shows the corresponding spectra for **2a** and **2b**, but not **2**. Authors should provide the spectrum using regioisomer mixture **2** to support their statement.

To facilitate a comparison of the data described in the manuscript, we have combined the protein mass spectra obtained upon purification with **2**, **2a**, and **2b** in one figure – see new Supplementary Fig. 5.

10b. Moreover, it will be helpful to show the spectra for the other two proteins (AqpZ-GFP and MATE) to support and generalize the statement. It will be interesting to compare the spectra for AmtB-MBP, where the difference between **2** and **2a**, **2b** is not as significant as that of AqpZ-GFP.

Following the referees suggestion we recorded mass spectra for AmtB-MBP upon purification with **2a** and **2b** and compared them with the spectrum obtained upon purification with the regioisomer mixture **2** (= **2a** + **2b**). We comment on the experimental outcome as follows: “As mentioned before in the case of AmtB, differences in extraction efficiency between **2**, **2a**, and **2b** were less pronounced. For all three OGD batches, mass spectra of comparable quality were obtained for AmtB (Supplementary Fig. 6). This demonstrates that the utility of OGDs for protein extraction is not necessarily limited to their regioisomer mixtures. If the targeted protein is sufficiently stable, individual OGD regioisomers can also be used for the purification and native MS analysis of membrane proteins.”

Moreover, we revised the results and discussion section of our manuscript to explain how the data obtained from MATE are supporting the conclusions presented in our manuscript – see comment to Reviewer #1, No. 8. #

11. In Page 9 line 7-10, authors mentioned that harsher activation is needed for removal of DDM as compared with OGDs. Authors refer to SI Figure 4 – 6 to support their statement. To my understanding, authors should refer to SI Figure 3 – 6 and specify the Collision voltages for the spectra in SI Figure 3 – 5 to support their statement.

We updated the figure references and specified the applied collision voltages within all relevant figure captions in order to support our statement.

12. In Page 9 line 13, authors mentioned the average charge state of OmpF is +18. It should be +19 from SI Figure 7. Similar conclusion on charge reduction can be reached from SI Figure 3a, which can be added to the discussion.

We thank the referee for raising this point. In the previous version of the manuscript we indeed incorrectly stated that the average charge state of OmpF is 18+ instead of 19+. In the revised version we corrected this and provide further information about how the average charge state was calculated (see Supplementary Fig. 17).

13. In Figure 4a, it will be helpful for the authors to show the result of DDM for comparison.

Following the referee's suggestion we added extraction data for DDM to Figure 4. We comment on the outcome of our experiment as follows: "During our analysis, DDM was therefore again used as a reference for comparison with our best performing [G1] and [G2] OGD regioisomer mixtures 1 and 4. Higher relative protein concentrations were obtained from the [G1] OGD regioisomer mixture 1, which highlights its utility for extracting large GPCR quantities from biological membranes (Fig. 4). [...] This is in line with the results presented above for bacterial membrane proteins and underlines the utility of OGDs for tuning membrane protein and lipid co-purification under the experimental conditions employed." The Source Data file was updated accordingly.

We thank the referee for this suggestion, because the experimental outcome clearly helped to underline the consistency of the conclusion presented in our manuscript. #

Minor Comments of Reviewer #1:

14. At the beginning of Page 4, authors should provide full name for "NTSR1" since it has not been defined before.

We thank the referee and implemented the suggested correction.

15. At the end of Page 5, IMAC should stand for immobilized metal (ion) affinity chromatography. Authors missed "affinity".

We thank the referee and implemented the suggested correction.

16. In SI Figure 1, there are ten columns in the gel picture, for which all the columns are properly labeled except the seventh. The 7th column was labeled as "-". Please specify/explain the label in the Figure caption.

The gel positions mentioned by the referee refer to protein samples that were purified with [G2]-triazole-C18. This detergent was used as a control and revealed that changing the linker between the [G2] head group and linear C18 tail does not improve protein extraction. Following the referee's advice, these bands are now labeled with an asterisk and an explanation was added to the figure caption of Supplementary Fig. 1.

17. In the last page, initials for author "Hsin-Yung Yen" is spelled incorrectly as "H.-Y.J." in "Author Contributions".

We thank the referee for this comment. The initials of "Hsin-Yung Yen" were changed to "H.-Y.Y." in the author contributions.

Major Comments of Reviewer #2

The paper by Umer et al addresses an important issue in membrane protein structural characterization. Detergent conditions need to be optimized on a per protein level which can be both a time consuming and seemingly random process at times given the variety of different classes of detergent. The application of a modular, tunable, detergent class for structural biology studies is of great interest. I think this manuscript is well suited to nature communications and its broad audience however a few additional controls should be included, as detailed below.

1a. The authors compare the total protein extraction of the OGD detergents to the conventional detergent DDM, and then go on to compare the amount of endogenous lipid retained between the best performing G1 and G2 OGD detergents. However, the authors present seemingly inconsistent information for DDM; "In contrast to the reference detergent DDM, resolved charge states were obtained for every membrane protein tested when using OGDs (Supplementary Fig. 4 – 6)." In SI Figure 6 an unresolved MS is shown. However, in SI Table 1 the authors give an average charge state for Aqpz-GFP and the mass of lipids binding in DDM. This suggests they must have been able to resolve the MS under some conditions. The authors should provide this MS. This must be included so that a full comparison between the different detergents can be made.

We were indeed not able to fully resolve mass spectra of other proteins upon purification with DDM under the employed experimental conditions. However, in the case of AqpZ, we could partially resolve at least three protein charge states and lipid-bound states. To highlight this data, we added an inset into the revised version of Supplementary Fig. 11 in which a zoom into the relevant m/z range of the spectrum is shown. We also realized that the spectral quality does not allow a proper calculation of average charge states. Therefore, we labeled the z_{ave} value listed for AqpZ-GFP and DDM in Supplementary Table 2 with an asterisk (*) and added the following explanation: "This value displays the most abundant charge state obtained from a partially resolved spectrum upon purification with DDM (see Supplementary Fig. 11)."

1b. The authors state that the native state of the protein is retained in these detergents. However, in Figure 2c only the m/z range containing the tetramer is shown. The m/z range should include the low mass range as this is needed to show that only the native oligomeric state is produced. The comparison to DDM mentioned above should also include the extended m/z range.

Following the referee's suggestion, we added mass spectra of AqpZ to the supplementary information which include also the low m/z range. In case of AqpZ, also dimers are obtained. In the revised version of the manuscript we refer to this observation as follows: "The mass spectra obtained following the isolation of AqpZ with [G1] OGD batches revealed a well-resolved tetrameric complex in its *apo* form (Fig. 2c). Therefore, we conclude that the oligomeric state of AqpZ was retained during isolation. In the lower mass range of the spectrum AqpZ dimers of lower intensity were observed (Supplementary Fig. 4). This suggests that OGDs are also capable of solubilizing partially assembled states of oligomeric AqpZ. Such partial assemblies are commonly removed by using further purification techniques, such as size-exclusion chromatography (SEC)³⁰. Mass spectra obtained from other bacterial membrane proteins, such as AmtB, MATE, and OmpF, show exclusively the expected oligomeric states (Supplementary Fig. 2, 5-9 and 12). In summary, our MS data highlight the utility of OGDs to preserve native oligomeric states of membrane proteins during purification."

2. For the charge reduction experiments the authors should include a comparison with the known charge reducing detergent C8E4.

Following the referee's suggestion we acquired a mass spectrum of OmpF in C8E4 and added the data to the revised supplementary information (see Supplementary Fig. 13). In the revised manuscript we refer to the experimental outcome as follows: "The z_{ave} obtained from the [G1] OGD regioisomer mixture **2** (15+) was lower than the value obtained from the charge-reducing detergent C8E4 (17.5+, Supplementary Fig. 12-13)."

3. For all detergents except OGD1, 2x the critical aggregation concentration (cac) is used. For OGD1 only 1x the cac was used. The authors should add an explanation for this into the SI.

We answer this question in the revised version of the supplementary information as follows (see Supplementary Table 4): "For MS experiments with [G1] OGD **1** the detergent concentration was adjusted to $1xcac$, because the cac of this detergent batch is considerably higher than the cac values of the other OGD batches (see Supplementary Table 3). However, further tests confirmed that adjusting the [G1] OGD **1** concentration to $2xcac$ does not affect the quality of the mass spectra. Source data are provided as a Source Data file."

4. For NTSR1, the authors only show extraction with OGDs and no comparison with DDM. However, they then go on to compare the binding activity of NTSR1 in DDM and OGD1. The authors should therefore also include the extraction efficiency in DDM. They also do not show any MS comparison with DDM. This should be included.

Following the referee's suggestion we added extraction data for DDM to Figure 4 – for further explanations see comments to Reviewer #1, No. 13. In the revised manuscript we refer to the MS comparison with DDM as follows: "Resolved mass spectra could not be obtained, however, following purification with DDM."

Minor Comments of Reviewer #2:

5. Abstract- “even though a large variety of detergents has been developed” should be “even though a large variety of detergents have been developed”

We thank the referee and implemented the suggested correction.

6. Page 4 line 1-define NTSR1 at its first use.

We thank the referee and implemented the suggested correction.

7a. Page 5-Given the broad readership of this journal I think it is essential to explain why tuning the charge states of membrane proteins in the MS is of great interest to the native MS community. In addition, describing what “favorable characteristics for native MS” are is essential.

We thank the reviewer for this very helpful comment. In the revised manuscript, we refer to the importance of tuning the charge states of membrane proteins as follows: “In general, low activation MS conditions and reduced protein charge states are favored when performing native MS. Both parameters help to preserve native structural features of membrane protein complexes, because they decrease the effect of Coulomb-driven protein unfolding during detergent removal inside the mass spectrometer^{20, 31}.”

Another favorable characteristic for MS is the ease of the detergent removal inside the mass spectrometer. We refer to this aspect in the revised manuscript as follows: “OGDs can be readily removed from proteomicelles by collisional activation inside the mass spectrometer, which facilitates the MS analysis of membrane protein-lipid complexes in general.” #

7b. What metrics are the authors using here to discern which detergents are the most favorable.

The utility of a detergent in membrane protein research varies with the application. This also makes it difficult to define metrics for a detergent that is equally suitable for all applications in membrane protein research. However, the OGD design guidelines deduced in our manuscript help to optimize the structure of detergents (= select the most favorable detergent) for individual applications in membrane protein research. We refer to this point in the revised manuscript as follows: “None of the OGDs investigated here is universally suitable for all applications in membrane protein research. However, OGDs present the first example of a detergent family whose molecular structures can be optimized for individual application in membrane protein research. This fine-tuning enables protein purification, preservation of lipid interactions during purification, facilitates MS analysis of membrane proteins and their lipid complexes, and confers protein charge reduction.”

Furthermore, in the revised manuscript, we describe individual metrics that allow us to tune the utility of OGDs for individual applications as follows: “

- 1.) “[...] tuning i) the heterogeneity of the OGD batches and ii) the structure of the OGD head and tail; are key parameters for optimizing the observable protein yields upon isolation.”

- 2.) “[...] relative protein amounts and lipid binding interactions can be controlled practically by tuning the structure of the OGD head and tail (Fig. 2c, Supplementary Fig. 2, 8-9).”
- 3.) “OGDs can be readily removed from proteomicelles by collisional activation inside the mass spectrometer [...]”
- 4.) “Finally, protein charge reduction can be tuned by varying the basicity of the linker between OGD head and tail.”

#

8. Page 5- “Different constructs of the aquaporin channel (AqpZ), the ammonia channel (AmtB), and the multidrug efflux pump (MATE) were extracted from *E. coli* membranes using 1 – 5 (Figure 2a, Supplementary Information)” Should this sentence be referencing the SI? If so what in the SI?

We thank the referee for raising this point, which we have addressed above – see comments to Reviewer #1, No. 2.

9. The mass range should be increased in all SI figures to better demonstrate that only the native oligomer is observed and to also show any detergent interference. In addition, I would suggest the authors keep their symbol size consistent in the figures- particularly SI Figure 3 as it stands currently the smaller symbols can be confused for monomers- as they appear to be one small pink triangle as opposed to the grey and pink larger symbol.

Following the referee’s suggestion we increased the m/z range in mass spectra of MATE-GFP (Supplementary Fig. 2), AqpZ-GFP (Supplementary Fig. 4), and OmpF (Supplementary Fig. 12). We did not increase the m/z range in Supplementary Fig. 5-9, because the chosen m/z range makes it easier to follow changes in protein charge states and lipid binding. Furthermore, we unified the size of the symbols within the spectra of AmtB to show that only the trimeric oligomeric state is obtained.

10. The authors should also label other high intensity peaks in the MS-e.g. in SI Figure 5 b bottom panel-please label the peaks between 6000-7000 m/z

We thank the referee for this comment and labeled the peaks between 6000 and 7000 m/z .

11. In the methods can the authors clarify if the same HCD energy was used for every MS acquired, if so state it in methods. If not it should be added to figure captions.

We describe the applied HCD energies in the methods section of our revised manuscript and also in all relevant figure captions.

Major Comments of Reviewer #3:

In the publication “Modular Detergents Tailor the Investigation of Membrane Proteins Including G-protein Coupled Receptors”, Urner et al. describe the synthesis of several oligoglycerol detergents (OGDs) and their utility for solubilising membrane proteins for purification and subsequent analysis by native mass spectrometry. The authors demonstrate the utility of these detergents for the solubilisation/mass spectrometric analysis of several integral membrane proteins, including the analysis of endogenous lipids carried through purification protocols. I have a number of concerns about the manuscript, including the level of interest of this work to the broad readership of Nature Communications, and suggest that revisions need to be made prior to publication to strengthen the evidence for the conclusions made. The work presented in this manuscript does not present a ‘silver bullet’ solution to the problem of detergent screening for native MS of membrane proteins, and whilst it is exciting that the authors have identified a new class of detergents capable of solubilising bacterial membrane proteins, there remains some gaps in the characterisation of their properties.

1a. The title of the manuscript is misleading and should be changed. What do the authors mean by ‘Tailor the Investigation’? The title suggests that the OGDs have been validated for use by many structural techniques, but the authors mostly present native mass spectrometry data (with some other limited solution based data). The title should reflect this.

The utility of detergents for membrane protein research varies with the application. The process of finding the right detergent for purification and structural analysis often appears to be random and can be time-consuming. By using the phrase “Modular Detergents Tailor the Investigation of Membrane Proteins” we want to indicate that we found a tunable detergent class, whose molecular structure can be optimized for the purification of membrane proteins and their analysis by native mass spectrometry and other biophysical assays. In this way, our detergents can “tailor (= customize)” the process of investigation. While reviewing the referee’s comment we come to the conclusion the word “Investigation” within the previous title was probably too general. Therefore, we rephrased the title as follows: “Modular Detergents Tailor the Purification and Analysis of Membrane Proteins Including G-Protein Coupled Receptors.”

1b. I also don’t see why the authors specifically mention G-protein coupled receptors in the title.

Given the fact that only a limited number of detergents is currently able to guide the extraction and structural analysis of G-protein coupled receptors, we anticipate that it is worth mentioning that our detergent family can also guide the purification and analysis of G-protein coupled receptors (GPCRs). This is currently one of the most challenging and most interesting protein families in pharmacology, because GPCRs are targets for more than 30% of all drugs on the market.

2. For AqpZ, AmtB and MATE purifications, membranes were solubilised with OGDs for 16 hours. Is there a reason why the solubilisation was so long? Have the authors tested the solubilisation efficiency as a function of time? Is this comparable with conventional detergents?

In the revised version of the manuscript we refer to the solubilization time as follows: “Following a previous purification protocol²⁵, cell membranes were solubilized for 16 hours and purified *via* immobilized metal ion affinity chromatography (IMAC).”

The solubilization time is comparable to the time used for conventional detergents. #

3. It is very confusing when the authors refer to “two OGD regioisomers” and a “mixture of detergent regioisomers”. This must be clarified throughout the manuscript. Perhaps Figure 1 could be improved in a way that more clearly depicts the structure and heterogeneity of the detergent mixtures used.

We thank the referee for suggesting to change the design of Figure 1. We have addressed this issue in one of the previous comments – see comment to Reviewer #1, No. 4.

Furthermore, we clarified the definition of OGD regioisomers and OGD regioisomer mixtures throughout the revised manuscript and added appropriate abbreviations to the relevant text passage: “Apparently, the [G1] OGD regioisomer mixture **2** (= **2a** + **2b**) is more suitable for the extraction and subsequent MS analysis of AqpZ than the individual [G1] OGD regioisomers **2a** and **2b**.” #

4. OGDs in no way resemble the heterogeneity and complexity of the biological membrane – references to this should be removed.

We have addressed this point in one of the previous comments – see comment to Reviewer #1, No. 4.

5. In Supplementary Figure 1, purification data for MATE are not shown.

We thank the referee for raising this point, which we have addressed in one of the previous comments – see comment to Reviewer #1, No. 7.

6. In Supplementary Figure 2, did the authors identify the lipids bound to MATE?

We identified the lipids bound to MATE and added the data to the Supplementary Table 2 and Source Data file.

7a. Are the in-solution stoichiometries identical for the proteins purified in each detergent? Are the samples homogeneous (e.g. using SEC-MALS)?

At the current stage of purification, e.g. upon IMAC, the in-solution stoichiometries are not expected to be homogenous. All possible oligomeric states of a protein would contain His-tags and are therefore not separable by IMAC under the experimental conditions employed. However, our MS data indicate that the solution stoichiometries obtained upon solubilization and IMAC purification of MATE, AmtB, OmpF, and NTSR1 are similar, because exclusively the expected oligomeric states were detected. Only in case of AqpZ, tetramers and dimers were detected. In the revised manuscript we comment on this observation as follows: “This suggests that OGDs are also capable of solubilizing partially assembled states of oligomeric AqpZ. Such partial assemblies are commonly removed by using further purification techniques, such as size-exclusion chromatography (SEC)³⁰.”

For the conclusion presented in our manuscript it is important that all samples are of comparable purity, which we have proven by means of SDS page analysis (see Supplementary Fig. 1-3). For those reasons, we feel that the SEC-MALS experiments suggested by the referee would not contribute to a deeper understanding of the conclusions made in our manuscript. #

7b. Are their structures identical (e.g. using CD)?

Data obtainable from CD spectroscopy display an average of all secondary structural elements that are present within a sample. Due to the low resolution nature of this method, it will be very difficult to prove that the secondary structures of all membrane proteins are identical within a sample.

However, our MS data show that OGDs can preserve the expected oligomeric states of membrane proteins during purification. The preservation of the native quaternary protein structure during purification also indirectly proves that native tertiary as well as secondary structural elements must have been preserved. This conclusion is further strengthened by K_d data which we have recorded for a NTSR1-agonist complex as part of the revision process (see comment to Reviewer #3, No. 13). The K_d values obtained from the NTSR1-agonist complex upon purification are in the nanomolar range, which is similar to data reported from cell-based assays. This underlines that OGDs can preserve significant proportions of native protein folds during purification. #

7c. Are the purified proteins (more) stable in the OGDs?

Following the referee's question, we investigated exemplarily the stability of MATE-GFP and AqpZ-GFP against precipitation in MS buffers containing [G1] OGD 1, [G2] OGD 4, or DDM. In the revised version of the manuscript we refer to the outcome of this experiment as follows: "The stabilities of both proteins against precipitation in MS buffer were similar in all three detergent environments. This further emphasizes the utility of OGDs for the structural analysis of membrane proteins by native MS (Supplementary Fig. 10)."

We added related experimental data together with a short description of the experiment to the revised supplementary information. The Source Data file was updated accordingly. #

7d. In Supplementary Figure 3, for example, varying amounts of trimeric and monomeric AmtB are detected by native MS, do these monomers result from gas-phase dissociation or are they also present in solution?

We unified the size of the AmtB symbols to clarify that only the trimeric oligomeric state of AmtB was obtained – see comment to Reviewer #2, No. 9.

8. On page 8 the authors state that poorly-resolved charge state distributions were observed for AqpZ upon extraction with individual regioisomers of 2a and 2b and attribute this to perturbation of the protein structure during isolation. Do the authors have any evidence for this? Couldn't this also be attributed to the individual regioisomers requiring additional activation to release them from the protein?

A further increase of the activation energy did not lead to an improvement of the protein mass spectra. This indicates that the protein-detergent compatibility may be more relevant for the quality of the mass spectra rather than the activation energy that is applied for the removal of the detergent micelle under the employed experimental conditions. In the revised manuscript, we no longer refer to the perturbation of the protein structure during isolation. Instead, we now state: "Apparently, the [G1] OGD regioisomer mixture **2** (= **2a** + **2b**) is more suitable for the extraction and subsequent MS analysis of AqpZ than the individual [G1] OGD regioisomers **2a** and **2b**."

9. Could the OGDs be used to solubilise OmpF directly from membranes? All other tested IMPs were α -helical, and demonstrating the ability of OGDs to solubilise β -barrel proteins would further emphasise their generic utility.

In the present manuscript we show that OGDs can solubilize membrane proteins from membranes and that OGDs can solubilize beta-barrel proteins upon detergent exchange. It is a very good suggestion to further emphasize the general utility of OGDs for protein purification. Instead of solubilizing beta-barrel proteins from membranes, we tested the utility of OGDs for the purification of beta-barrel proteins from inclusion bodies, because this marks an important alternative purification strategy for beta-barrel membrane proteins. In the revised manuscript we refer to the outcome of our additional experiment as follows: "Apart from alpha-helical membrane proteins, such as AqpZ, AmtB, and MATE, we could also purify the beta-barrel outer membrane protein T (OmpT) from inclusion bodies with DDM as well as [G1] and [G2] OGD regioisomer mixtures. Again, higher relative protein quantities were obtained from the [G1] OGD regioisomer mixture **1** (Supplementary Fig. 3). [...] Our data indicate that the structural impact of OGD head groups and tails on protein purification can be extrapolated to alpha-helical and beta-barrel proteins." We thank the referee for encouraging us to perform this experiment as it clearly helped to emphasize the general utility of OGDs for the purification of membrane proteins.

10a. Does charge reduction alter the apparent affinities of the protein-ligand interactions detected by MS? Is the increased lipid observed in some OGDs due to the protein-lipid complexes being stabilised in different detergents in the gas phase (or in solution too)?

To answer these questions we added another paragraph to the revised results and discussion section of our manuscript: "An important question prompted by this study is related to the increase in relative intensities of protein-lipid complexes detected by MS. Is this increase due to their stabilization in solution or in the gas phase? To answer this question we again focus on protein mass spectra obtained upon purification with [G1] and [G2] OGD regioisomer mixtures **1** and **4**. The spectra obtained for tetrameric AqpZ, for example, show similar protein charge states (Fig. 2c). Both detergents are not charge-reducing and the ionized protein-complexes experience similar collisional activation during detergent removal inside of the mass spectrometer²⁹. Higher relative intensities of protein-lipid complexes were detected when the [G2] OGD regioisomer mixture **4** was used. The [G2] OGD regioisomer mixture **4** exhibits larger head groups than mixture **1**, as well as a more lipid-like tail (Fig. 2c). Furthermore, protein mass spectra obtained from [G1] and [G2] OGD regioisomer mixtures **2** and **5** confirm this observation. Both detergent batches reduce the charge of membrane proteins thus leading to softer collisional activation conditions within the HCD cell under the experimental conditions employed.

However, protein-lipid complexes were again only obtained upon purification with [G2] OGD regioisomer mixture **5**, which exhibits larger head groups and a more lipid-like tail (Supplementary Fig. 9). This leads us to the conclusion that the increase in relative intensities of protein-lipid complexes depend more on their stabilization by the proteomicelle in solution rather than on charge effects in the gas phase.” #

10b. Have the authors used any other methods to quantify the lipids extracted with the membrane proteins, apart from native MS?

In our manuscript we investigate how the preservation of protein-lipid interactions in proteomicelles can be optimized by tuning the structure of the detergent. Native MS allows us to investigate specifically those lipids that remain bound to proteins after removal of the detergent micelle. In contrast, colorimetric- and chromatography-based lipid quantification assays provide average information about the entire lipid population, which is spread over protein-free detergent micelles and proteomicelles in solution. For those reasons, we feel that other lipid quantification methods are unlikely to yield a meaningful outcome.

10c. The authors have previously implemented an OmpF-peptide binding experiment (e.g. Housden et al., Science, 2013). Are the measured affinities in the OGDs different than OG, and do they differ between detergents? The same question could be asked for apparent affinities measured for lipids titrated into the OGD-solubilised IMPs.

In our present manuscript we show how the molecular structure of OGDs can be optimized for individual applications in membrane protein research, including protein purification, preservation of lipid interactions during purification, easy MS analysis of membrane protein complexes, and protein charge reduction. For those reasons we feel that the experiments suggested by the referee would not contribute to a deeper understanding of the statements made in the manuscript rendering further OmpF-peptide binding or lipid titration experiments not necessary at this point.

11. What yield is obtained in DDM relative to OGDs for NTSR1? The comparison presented in Figure 4a doesn't appear to be relative to DDM extraction, and in Supplementary Table 1 data for only detergent 1 is shown, whereas detergent 4 was also used to solubilise NTSR1 (Fig 4a), and DDM purified protein was used in binding assays as a comparison (Supplementary Figure 8). Additionally, were multiple biological replicates performed as error bars are not shown like in other figures?

The requested experimental data including error bars were added to Figure 4 and three biological replicates were performed. In the revised manuscript, we refer to the experimental outcome as follows: “During our analysis, DDM was therefore again used as a reference for comparison with our best performing [G1] and [G2] OGD regioisomer mixtures **1** and **4**. Higher relative protein concentrations were obtained from the [G1] OGD regioisomer mixture **1**, which highlights its utility for extracting large GPCR quantities from biological membranes (Fig. 4).”

12. On page 11, the authors state “the finding that high relative intensities of lipid complexes were obtained upon isolation with 1 is not surprising, since it is known that lowering the extraction time effectively lowers protein delipidation”. If that is the case, do lower extraction times with G1 OGDs for the other proteins studied result in increased lipid being retained throughout purifications? If G2 OGDs are less effective at solubilising membranes then I presume the effective extraction time is reduced (as it takes longer to solubilise proteins from the bilayer) so this could explain why the authors see increased amounts of bound lipid with these detergents.

We thank the referee for raising this point because it has clearly helped to strengthen the consistency of the conclusions presented in our manuscript. In our initial experiments, the extraction time used for MBP-NTSR1-TrxA was reduced from 16 h to two, because in comparison to bacterial membrane proteins, GPCRs are more prone to denaturation in the presence of detergents. Following our protocol, it was difficult to prove that [G2] OGD 4 preserves more lipid binding than [G1] OGD 1, because NTSR1 precipitated in 4 after proteolytic removal of the MBP and TrxA tags. Since it is known that the extraction time can affect the extent of delipidation, we previously concluded that the protein-lipid complexes detected upon purification with [G1] OGD 1 must have been a result of the reduced extraction time.

During the revision process, we successfully purified MBP-NTSR1-TrxA by IMAC after extending the solubilization time from two to 16 h. In contrast to NTSR1, the MBP-NTSR1-TrxA construct is stable in both OGDs 1 and 4. We analyzed MBP-NTSR1-TrxA directly upon IMAC by native MS. In the revised manuscript we refer to the outcome of this experiment as follows: “Native MS experiments revealed reduced relative intensities of protein-lipid complexes upon purification with the [G1] OGD regioisomer mixture 1. Enhanced relative intensities of lipid complexes were observed in mass spectra following purification with [G2] OGD regioisomer mixture 4 (Supplementary Fig. 14). [...] This is in line with the results presented above for bacterial membrane proteins and underlines the utility of OGDs for tuning membrane protein and lipid co-purification under the experimental conditions employed.”

In the revised manuscript we no longer refer to the reduced extraction time, because the extraction time has been adjusted to the conditions used for bacterial membrane proteins. We removed the MS data of NTSR1 from Figure 4 and updated the Supporting Information with new MS data obtained from MBP-NTSR1-TrxA upon purification with OGDs 1 and 4 (see Supplementary Fig. 14). #

13. The data presented for agonist binding is weak. Could the authors use their fluorescein labelled peptides to determine K_d (app) values, for example by fluorescence polarisation? Is the affinity perturbed in OGDs?

Following the referee’s suggestion, we investigated the K_d values for the complex formed by NTSR1 and the dye-labeled agonist using affinity MS and fluorescence polarization. We refer to the outcome of our experiment in the revised version of the manuscript as follows: “Further affinity MS experiments revealed that the chromophore-labeled agonist binds strongly to NTSR1, when the protein was purified with the [G1] OGD regioisomer mixture 1. The K_d value obtained from the NTSR1-agonist complex is in the nanomolar range and similar to data reported from cell-based assays (Fig. 4)⁴⁵. Fluorescence polarization experiments support this result (Supplementary Fig. 16). This underlines that the [G1] OGD regioisomer mixture 1 preserves functional characteristics of the receptor during purification⁴⁶.”

We thank the referee for raising this point, which clearly helped to emphasize the utility of OGDs for protein purification. #

14a. Could the authors comment on the ratio of regioisomers chosen in Supplementary Table 2? Were these ratios selected in any particular way via the synthetic route?

We answer these questions in the revised version of the Supplementary Table heading as follows: “The regioisomer ratios result from the starting material that is used for the synthesis of OGDs. Regioisomer ratios determined for final products and intermediates are given in the synthetic procedures. Further information about the analysis of regioisomer ratios is given in the sub-chapter *Quantification of Regioisomer Proportions.*”

14b. Could these be tuned to give more favourable properties?

We answer this question in the manuscript as follows: “Apparently, the [G1] OGD regioisomer mixture **2** (= **2a** + **2b**) is more suitable for the extraction and subsequent MS analysis of AqpZ than the individual [G1] OGD regioisomers **2a** and **2b**. As mentioned before in the case of AmtB, differences in extraction efficiency between **2**, **2a**, and **2b** were less pronounced. For all three OGD batches, mass spectra of comparable quality were obtained for AmtB (Supplementary Fig. 6). This demonstrates that the utility of OGDs for protein extraction is not necessarily limited to their regioisomer mixtures. If the targeted protein is sufficiently stable, individual OGD regioisomers can also be used for the purification and native MS analysis of membrane proteins. The ability to optimize the performance of OGDs for protein purification by changing the regioisomer ratios depends on the targeted protein.”

15. In Supplementary Table 3, why was only 1xcac used for ODG 1, whereas 2x cac amounts were used for the other detergents?

We thank the referee for raising this point and added an explanation to the revised supplementary information: “For MS experiments with [G1] OGD **1** the detergent concentration was adjusted to 1xcac, because the *cac* of this detergent batch is considerably higher than the *cac* values of the other OGD batches (Supplementary Table 3). However, we found that adjusting the [G1] OGD **1** concentration to 2xcac does not affect the quality of the mass spectra.”

16. Did the authors employ smoothing and background subtraction during data analysis? In some instances signal-to-noise is poor, could the authors comment on why this is the case?

In the revised manuscript we comment on smoothing and background subtraction as follows: “UniDec was used for background subtraction and smoothing⁴⁹.” Differences in detergent compatibility are relevant for variations in mass spectral quality – see comment to Reviewer #3, No. 8.

17. The manuscript doesn't really have a discussion section. The authors should think about putting their work a bit in context with the literature more clearly. What sort of mass spectrometry experiments might it be useful to try retain lipid binding events? Why is it beneficial to not have to perform detergent exchange? What other detergents or purification systems for membrane proteins have been investigated, why are OGDs better? What other applications might benefit from custom detergents, and how?

We thank the reviewer for this helpful comment. We have substantially revised and extended the results and discussion section of the manuscript. In addition to that, we added two more paragraphs in order to address the questions raised by the referee:

“Detergents exhibit individual delipidation properties and protein delipidation can also change with the time that is used to expose membrane protein-lipid complexes to detergents^{32, 33}. From our experience, detergents that co-purify substantial amounts of lipids are often not suitable for the straightforward MS analysis of membrane protein complexes. DDM, for example, is known to co-purify substantial amounts of lipids and requires harsh MS activation conditions to achieve sufficient detergent removal^{32, 34}, which can hamper the detection of intact membrane protein complexes by MS³⁵. In practice, the investigation of protein-lipid interaction by MS is addressed mainly in two ways: First, membrane proteins are delipidated step-wise with detergents that exhibit weak delipidating properties, such as DDM. To do so, protein-lipid complexes are repetitively purified by SEC, IMAC, or dialysis until mass spectra of sufficient quality are obtained^{33, 36}. This allows us to investigate membrane proteins in complex with co-purified membrane lipids³⁷. In the second approach, membrane proteins are purified with detergents that exhibit strong delipidating properties, such as C8E4, OG, or LDAO^{25, 29}. If the targeted protein is sufficiently stable after delipidation, individual lipids can be added back to the sample solution^{25, 27}. Subsequently, MS analysis, gas-phase unfolding protocols, or functional assays allow us to study how the molecular structure of individual lipids affects the structure and function of membrane proteins^{25, 27, 38, 39}.”

In contrast to the above-mentioned methodologies, OGDs enable the straightforward analysis of interactions between membrane proteins and native membrane lipids directly after protein extraction and IMAC. Moreover, following our purification protocol, relative protein amounts and lipid binding interactions can be controlled practically by tuning the structure of the OGD head and tail (Fig. 2c, Supplementary Fig. 2, 8-9). This facilitates experimental access to either membrane proteins in complex with native membrane lipids or delipidated membrane proteins. In addition, OGDs can be readily removed from proteomicelles by collisional activation inside the mass spectrometer, which facilitates the MS analysis of membrane protein-lipid complexes in general. Finally, protein charge reduction can be tuned by varying the basicity of the linker between OGD head and tail. The ability to optimize the structure of OGDs for the purification and native MS analysis of protein-lipid interactions strengthens our anticipation that OGDs will facilitate the investigation of challenging membrane proteins in the future.” #

Minor Comments of Reviewer #3:

18. Bottom line of p3 – Remove the word “Remarkably”.

We thank the referee and implemented the suggested correction.

19. Four lines from bottom of p4 – Replace the word “outstanding”

We thank the referee and implemented the suggested correction.

20. Why is it of interest to tune the charge states of membrane proteins in mass spectra? Could the authors expand on this statement?

We thank the referee for raising this point, which we have addressed in one of the previous comments – see comment to Reviewer #2, No. 7a.

21. Change “comparative instrumental conditions” to “comparable instrumental conditions”

We thank the referee and implemented the suggested correction.

Reviewers' Comments:

Reviewer #1:

Remarks to the Author:

The authors nicely addressed all my concerns and comments in the revision. I would recommend the manuscript to be published in Nat. Commun. as is.

Another minor point, however, needs some attention. In the Supporting information, line 97 and 100, the caption should be "... batches 2, 2a, and 2b. ..." rather than "... batches 2, 2a, and ab. ...". Please correct it in the final round.

Reviewer #2:

Remarks to the Author:

The authors have addressed the concerns of the reviewers and I think the manuscript is suitable for publication in nature communications. The revised manuscript is clear and well written. There are a couple of minor typos and comments that should be addressed and are detailed below.

Given that a significant proportion of dimer can be observed for AqpZ in G1 OGD 1 and more so in G2 OGD 4 (SI figure 4) I think this should be shown in the main manuscript (Figure 2c) rather than in the SI. Although the authors mention this in the main manuscript it would be much clearer if shown in the figure in the main body of the paper. Especially as the authors assert that the native state is preserved (when in fact a significant proportion of dimer is also observed). Even with increasing the mass range the lipid binding for G2 OGD4 will be clear, especially with the inset provided.

Page 6 second line from the bottom of the page- "In case of AqpZ" should read "In the case of AqpZ"

Page 7 line 4- "We found similar trends in case of AmtB" should read "We found similar trends in the case of AmtB"

Supplementary information Figure 10 caption "span down" should be "spun down"

In supplementary information figure 15, the error bars are significantly higher for OGD1 than DDM. The authors should comment on this in the manuscript.

Reviewer #3:

Remarks to the Author:

In this revised manuscript Urner et al. have described the development and utility of a new class of detergents for the mass spectrometric analysis of membrane proteins. Whilst the authors have addressed some of my concerns in this revision, several issues remain that, in my view, must be addressed before publication.

- I still feel that the title remains too general and is misleading. The authors have only analysed their purified proteins by native MS and the title should reflect this. Indeed, most of the manuscript describes how the properties of the detergents can be tuned for MS analysis (charge reduction etc).
- The comment by the authors that they could "purify the beta-barrel outer membrane protein T (OmpT) from inclusion bodies" is incorrect. The protein has already been purified, and the authors are attempting to refold OmpT from its denatured state. I presume the protein amounts quantified in Supp Fig 3a were just based on soluble protein (determined by absorbance measurements), which is a poor measure of folding success – as their own data attests. In many cases they have not been successful in refolding the protein as judged by the intensities of the folded and unfolded

bands on the SDS-PAGE gel (where I presume the samples were not boiled prior to running on the gel – this detail is lacking) (Supp Fig 3). What is the difference in folding yield between DDM and the OGDs (as determined by quantifying the relative intensities of the folded and unfolded bands in b)? It appears to me that all the OGDs are less efficient than DDM at refolding the OMP under the conditions used. Was the refolded protein functional? This could be determined easily for this protein, for example by using an assay for OmpT protease activity (Kramer et al., *Eur. J. Biochem.* 2000, 267, 885–893). Could the authors attain spectra of OmpT? In my previous comments I was asking if the authors had tried purifying OmpF directly from membranes using OGDs.

- In Supplementary Figures 6 and 8, why are two Gaussian distributions for the trimeric complexes seen in the spectra? This seems very unusual, and suggests at least partial protein unfolding in the OGDs, despite preserving the expected oligomeric states of the proteins. Is this a solution or gas phase effect? Did the authors try to interrogate the solution properties of these detergent solubilised proteins? E.g. CD can be used to provide an easy indication of sample quality and batch-to-batch variability and I struggle to see why the authors haven't attempted these sort of standard quality control experiments given their MS data.
- The stability data presented in Supplementary Fig 10 aren't really what I was looking for. My question is how stable are the membrane proteins purified in OGDs? Are they stable enough for long-term storage?
- Given how keen the authors are to state the utility of their detergents for native MS of GPCRs, why are the corresponding spectra in the supplementary information and not the main text?
- Could the authors obtain a mass spectrum NTSR1 after removal of the MBP tag?
- The attached Reporting Summary and Editorial Policy Checklist are not prepared by the corresponding author of this paper.

In detail, we respond to the reviewers comments as follows:

Reviewer #1 (Remarks to the Author):

The authors nicely addressed all my concerns and comments in the revision. I would recommend the manuscript to be published in Nat. Commun. as is.

1.) Another minor point, however, needs some attention. In the Supporting information, line 97 and 100, the caption should be "... batches 2, 2a, and 2b. ..." rather than "... batches 2, 2a, and ab. ...". Please correct it in the final round.

We thank the referee and implemented the suggested correction.

Reviewer #2 (Remarks to the Author):

The authors have addressed the concerns of the reviewers and I think the manuscript is suitable for publication in nature communications. The revised manuscript is clear and well written. There are a couple of minor typos and comments that should be addressed and are detailed below.

1.) Given that a significant proportion of dimer can be observed for AqpZ in G1 OGD 1 and more so in G2 OGD 4 (SI figure 4) I think this should be shown in the main manuscript (Figure 2c) rather than in the SI. Although the authors mention this in the main manuscript it would be much clearer if shown in the figure in the main body of the paper. Especially as the authors assert that the native state is preserved (when in fact a significant proportion of dimer is also observed). Even with increasing the mass range the lipid binding for G2 OGD4 will be clear, especially with the inset provided.

This is a very good suggestion which we tried to implement, but it turned out to be difficult to include the extended mass spectra into Figure 2c without significantly increasing the size of Figure 2. At first glance, the relative intensity of the AqpZ tetramer looks indeed drastically lower in the case of [G2] OGD regioisomer mixture 4. However, the intensity of the tetramer is dispersed over its lipid-free form and lipid-bound states. These signals together contribute to the relative intensity of the tetramer. Therefore, when calculating the relative MS intensities of dimer and tetramer it becomes apparent that they are not drastically different between both detergents. We have clarified this in the revised figure caption of Supplementary Fig. 4 as follows: "The relative MS intensities of dimer and tetramer are 5:95 in the case of [G1] OGD regioisomer mixtures 1 and 15:85 in the case of [G2] OGD regioisomer mixture 4."

2.) Page 6 second line from the bottom of the page- "In case of AqpZ" should read "In the case of AqpZ"

We thank the referee and implemented the suggested correction.

3.) Page 7 line 4- "We found similar trends in case of AmtB" should read "We found similar trends in the case of AmtB"

We thank the referee and implemented the suggested correction.

4.) Supplementary information Figure 10 caption “span down” should be “spun down”

We thank the referee and implemented the suggested correction.

5.) In supplementary information figure 15, the error bars are significantly higher for OGD1 than DDM. The authors should comment on this in the manuscript.

In the revised supplementary information, we refer to this as follows: “In the case of OGD regioisomer mixture 1 the error bars are comparatively large due to an outlier (absorbance at 495 nm ~ 0.6, see overlaid dot blot). It is possible that this sample became less diluted than the others due to variations in Bio-Spin column handling or batch to batch variation of column properties.”

Reviewer #3 (Remarks to the Author):

In this revised manuscript Urner et al. have described the development and utility of a new class of detergents for the mass spectrometric analysis of membrane proteins. Whilst the authors have addressed some of my concerns in this revision, several issues remain that, in my view, must be addressed before publication.

1.) I still feel that the title remains too general and is misleading. The authors have only analysed their purified proteins by native MS and the title should reflect this. Indeed, most of the manuscript describes how the properties of the detergents can be tuned for MS analysis (charge reduction etc).

In our manuscript, we describe the utility of OGDs for the isolation and structural analysis of membrane proteins. Apart from detergent properties that are favorable for MS, such as easy detergent removal and charge reduction, we also investigate oligomeric states of bacterial membrane proteins and show how the structure of OGDs can be changed to optimize protein extraction and the preservation of lipid interactions during purification. Moreover, we show that the OGD regioisomer mixture 1 can purify a functional GPCR. In light of these results, we believe that our title is not misleading as our data show how changing the structure of OGDs allows us to tailor the purification and structural analysis of membrane proteins, including GPCRs. While reviewing the referees comment we conclude that the phrase “Modular Detergents Tailor the Purification and Analysis ...” was probably too general. Therefore, we rephrased the title as follows: “Modular Detergents Tailor the Purification and Structural Analysis of Membrane Proteins Including G-protein Coupled Receptors.” Furthermore, we specify the role of native MS for our work within the abstract.

2a.) The comment by the authors that they could “purify the beta-barrel outer membrane protein T (OmpT) from inclusion bodies” is incorrect. The protein has already been purified, and the authors are attempting to refold OmpT from its denatured state.

We thank the referee for raising this point as it helps to clarify the explanation of our experiments. During isolation of OmpT from inclusion bodies, the protein was first solubilized in urea (8 M) and then refolded by diluting the OmpT-urea solution into detergent-containing refolding buffer. After removing urea by means of IMAC, the protein is only water-soluble because of the detergent. In this way, detergents enable the solubilization of OmpT. We have clarified these aspects in the revised manuscript as follows: “[...] we also attempted to refold and solubilize the G236K/K237G mutant

of the beta-barrel outer membrane protein T (OmpT) with DDM as well as [G1] and [G2] OGD regioisomer mixtures. We first solubilized OmpT from inclusion bodies with urea, then diluted the OmpT-urea mixture into detergent-containing refolding buffer, and isolated the protein using IMAC.” #

2b.) I presume the protein amounts quantified in Supp Fig 3a were just based on soluble protein (determined by absorbance measurements), which is a poor measure of folding success – as their own data attests. In many cases they have not been successful in refolding the protein as judged by the intensities of the folded and unfolded bands on the SDS-PAGE gel (where I presume the samples were not boiled prior to running on the gel – this detail is lacking) (Supp Fig 3). What is the difference in folding yield between DDM and the OGDs (as determined by quantifying the relative intensities of the folded and unfolded bands in b)? It appears to me that all the OGDs are less efficient than DDM at refolding the OMP under the conditions used.

We thank the reviewer for this helpful comment, which helped to improve the clarity of our data interpretation. We specified the proportion of folded and unfolded OmpT determined by SDS page analysis in the revised version of Supplementary Fig. 3 and clarified that the samples were not boiled prior SDS page analysis. Furthermore, we refer to refolding efficiency of DDM and OGDs in the revised manuscript as follows: “Analysis of the ratio between folded and unfolded OmpT revealed higher relative proportions of folded OmpT in the case of DDM (Supplementary Fig. 3). However, relative protein quantities obtained from the [G1] OGD regioisomer mixture 1 were about three times higher than from DDM, thus leading to a higher absolute amount of refolded OmpT under comparable purification conditions (Supplementary Fig. 3).”

2c.) Could the authors attain spectra of OmpT?

Following the referees question we acquired a mass spectrum of OmpT upon purification with [G1] OGD regioisomer mixture 1. The average charge state is similar to the most abundant charge state obtained for DDM in the literature, which is a detergent that is not associated with charge reduction. This is fully consistent with the data obtained from OmpF and underlines that the [G1] OGD regioisomer mixture 1 exhibits no charge reducing properties under the experimental conditions employed. We added the spectrum to the supporting information and refer to the experimental outcome as follows: “Further analysis of a mass spectrum obtained from OmpT upon purification with [G1] OGD regioisomer mixture 1 supported this conclusion. The Z_{ave} of OmpT obtained from 1 (11+) is similar to the most abundant protein charge state obtained from DDM (10+)³¹, a detergent that is also not associated with charge reduction (Supplementary Fig. 13)²⁹.”

2d.) Was the refolded protein functional? This could be determined easily for this protein, for example by using an assay for OmpT protease activity (Kramer et al., Eur. J. Biochem. 2000, 267, 885–893). In my previous comments I was asking if the authors had tried purifying OmpF directly from membranes using OGDs.

We thank the referee for raising this point. We attempted to refold the purified OmpT from its denatured state in urea, which is now clarified in the revised manuscript – see reply to Reviewer #3, No. 2b. Our aim was to evaluate how efficient OGDs are in solubilizing and refolding OmpT. We used the OmpT variant carrying

the mutations G236K/K237G as described before by Kramer and co-workers: <https://doi.org/10.1046/j.1432-1327.2000.01073.x>. Given the fact that our OGDs do not enable quantitative refolding of OmpT future experiments, beyond the scope of this manuscript, will be directed towards improving the efficiency of OGDs in refolding beta-barrel membrane proteins, including also a functional characterization.

3.) In Supplementary Figures 6 and 8, why are two Gaussian distributions for the trimeric complexes seen in the spectra? This seems very unusual, and suggests at least partial protein unfolding in the OGDs, despite preserving the expected oligomeric states of the proteins. Is this a solution or gas phase effect? Did the authors try to interrogate the solution properties of these detergent solubilised proteins? E.g. CD can be used to provide an easy indication of sample quality and batch-to-batch variability and I struggle to see why the authors haven't attempted these sort of standard quality control experiments given their MS data.

The appearance of more than one charge state distribution is well-known in protein mass spectrometry. In the case of membrane proteins, at least two explanations are available. We comment on them in the figure captions of the revised supplementary information as follows: "Bimodal charge state distributions are likely a consequence of charge-stripping effects or partial unfolding of the MBP tags. For further information about charge-stripping effects see Reading *et al.*²"

Further investigation of these phenomena is beyond the scope of our paper since it will not contribute to a deeper understanding of the conclusions drawn from these figures. #

4.) The stability data presented in Supplementary Fig 10 aren't really what I was looking for. My question is how stable are the membrane proteins purified in OGDs? Are they stable enough for long-term storage?

We thank the referee for raising this point, because it helps to underline the utility of OGDs for the structural analysis of membrane proteins. In the revised manuscript, we refer to the experimental outcome as follows: "Moreover, the isolated proteins were stable in solution and could be analyzed by native MS even after multiple freeze-thaw cycles. This further emphasizes the general utility of OGDs for the structural analysis of membrane proteins by native MS."

5.) Given how keen the authors are to state the utility of their detergents for native MS of GPCRs, why are the corresponding spectra in the supplementary information and not the main text?

We thank the referee for encouraging us to transfer our GPCR MS data from the supplementary information to Figure 4 in the manuscript. The key information provided in this figure is that lipid preservation during protein purification can be controlled by changing the structure of the head group and tail of OGDs. In this way, our GPCR MS data are supporting the main conclusion which we have drawn from the data presented before in Figure 2. Therefore, we prefer to leave our GPCR MS data within the supplementary information.

6.) Could the authors obtain a mass spectrum NTSR1 after removal of the MBP tag?
The MBP and ThrxA tags are sensitive to the same protease rendering a selective removal of the MBP tag difficult.

7.) The attached Reporting Summary and Editorial Policy Checklist are not prepared by the corresponding author of this paper.

We forwarded this information to the editorial office.

Reviewers' Comments:

Reviewer #2:

Remarks to the Author:

The authors have addressed all of my comments and I think the manuscript is ready, and suitable, for publication in Nature Communications.

Reviewer #3:

Remarks to the Author:

The authors have thoroughly addressed all my concerns, and I recommend publication of the manuscript in Nature Communications. I couldn't find a reference in the text to Supplementary Fig. 19, so this should be double checked.

Reviewers' Comments:

Reviewer #2:

Remarks to the Author:

The authors have addressed all of my comments and I think the manuscript is ready, and suitable, for publication in Nature Communications.

Reviewer #3:

Remarks to the Author:

The authors have thoroughly addressed all my concerns, and I recommend publication of the manuscript in Nature Communications. I couldn't find a reference in the text to Supplementary Fig. 19, so this should be double checked.

We thank the referee and included a reference to Supplementary Figure 19 in the text of the revised manuscript.